# Modeling single-cell phenotypes links yeast stress acclimation to transcriptional repression and pre-stress cellular states

**Andrew C Bergen[1†], Rachel A Kocik[1†], James Hose[1], Megan N McClean[1,2,3], Audrey P Gasch[1,3,4]\***

[1]Center for Genomic Science Innovation, University of Wisconsin-Madison, Madison, United States; [2]Department of Biomedical Engineering, University of Wisconsin-Madison, Madison, United States; [3]University of Wisconsin Carbone Cancer Center, University of Wisconsin School of Medicine and Public Health, Madison, United States; [4]Department of Medical Genetics, University of Wisconsin-Madison, Madison, United States

**\*For correspondence:**
agasch@wisc.edu

[†]These authors contributed equally to this work

**Competing interest:** The authors declare that no competing interests exist.

**Abstract** Stress defense and cell growth are inversely related in bulk culture analyses; however, these studies miss substantial cell-to-cell heterogeneity, thus obscuring true phenotypic relationships. Here, we devised a microfluidics system to characterize multiple phenotypes in single yeast cells over time before, during, and after salt stress. The system measured cell and colony size, growth rate, and cell-cycle phase along with nuclear trans-localization of two transcription factors: stress-activated Msn2 that regulates defense genes and Dot6 that represses ribosome biogenesis genes during an active stress response. By tracking cells dynamically, we discovered unexpected discordance between Msn2 and Dot6 behavior that revealed subpopulations of cells with distinct growth properties. Surprisingly, post-stress growth recovery was positively corelated with activation of the Dot6 repressor. In contrast, cells lacking Dot6 displayed slower growth acclimation, even though they grow normally in the absence of stress. We show that wild-type cells with a larger Dot6 response display faster production of Msn2-regulated Ctt1 protein, separable from the contribution of Msn2. These results are consistent with the model that transcriptional repression during acute stress in yeast provides a protective response, likely by redirecting translational capacity to induced transcripts.

## Editor's evaluation

This paper addresses an important question in the field: the cell-to-cell heterogeneity in stress response and the functional relevance to stress adaptation. The experimental approaches are timely and their clustering and correlation analyses suggest some interesting relationships between phenotypic factors and growth adaptation.

## Introduction

All organisms respond to cellular stress, which can arise from external conditions such as drugs and environmental shifts or internal perturbations including mutation and disease. Thus, at the cellular level, organisms must be able to sense both external and internal signals to mount a proper response. Yet in both single- and multi-celled organisms, there can be large variation in how individual cells respond to environmental stress, even among genetically identical cells in the same environment. For example, cell-to-cell variation in signaling and gene expression have been linked to differential

survival of isogenic cancer cells responding to drugs (*Lee et al., 2014*; *Paek et al., 2016*; *Shaffer et al., 2017*; *Inde and Dixon, 2018*). Similarly, cellular heterogeneity in bacterial growth and gene expression can produce variation in survival upon antibiotic treatment (*Balaban et al., 2004*; *Keren et al., 2004*). Understanding the nature of this variation could facilitate the modulation of stress survival, with therapeutic applications.

One marker of heterogeneity in stress responses is dynamic localization of stress-activated transcription factors. Several canonical factors, including p53 in mammalian cells (*Purvis et al., 2012*; *Kracikova et al., 2013*; *Paek et al., 2016*) and Msn2 and its paralog Msn4 in fungi (*Görner et al., 1998*), reside in the cytosol in the absence of stress but rapidly translocate to the nucleus upon activation. These and other stress-activated factors can vary substantially in their responsiveness, in ways that can impact cellular outputs including gene-expression. For example, Msn2 localization dynamics differ depending on the nature of the stress (*Hao and O'Shea, 2012*; *Petrenko et al., 2013*; *Granados et al., 2018*), and these differences impart distinct effects on different target genes (*Hao and O'Shea, 2012*; *Hansen and O'Shea, 2013*; *Stewart-Ornstein et al., 2013*; *Hansen and O'Shea, 2015a*; *Hansen and O'Shea, 2015b*; *Hansen and O'Shea, 2016*; *Hansen and Zechner, 2021*). Msn2 targets with highly responsive promoters can be induced even with brief pulses of nuclear Msn2, whereas genes with less responsive promoters require prolonged Msn2 activation (*Hansen and O'Shea, 2013*; *Hansen and O'Shea, 2015a*; *Hansen and O'Shea, 2015b*; *Hansen and O'Shea, 2016*). Similarly, differences in the dynamics of p53 localization can lead to distinct transcriptional outputs, and these distinctions correlate with differences in stress survival (*Purvis et al., 2012*). Several studies have observed substantial cell-to-cell heterogeneity in nuclear localization dynamics of these factors (*Cai et al., 2008*; *Cheong et al., 2011*; *Purvis and Lahav, 2013*; *Lin et al., 2015*; *AkhavanAghdam et al., 2016*; *Gasch et al., 2017*; *Granados et al., 2018*; *Li et al., 2018*); however, the causes and functional effects of this variation remain poorly understood.

Cell-to-cell variation in transcription factor localization dynamics could arise for several reasons. Changes in the state of a single transcription factor may alter its localization independent of or

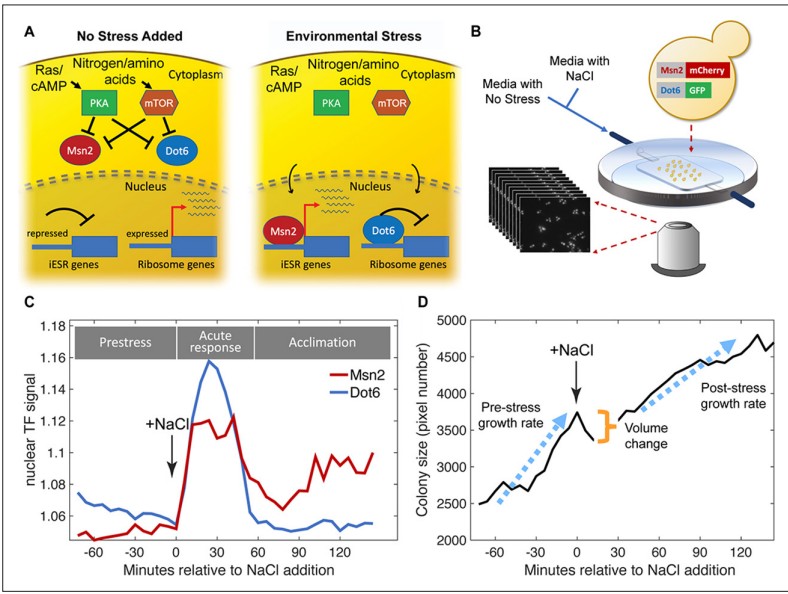

**Figure 1.** Experimental approach. (**A**) Schematic of Msn2 and Dot6 localization in the absence (left) and presence (right) of stress. (**B**) Diagram of microfluidic device used for time-lapse microscopy. (**C**) Representative nuclear localization scores (see Methods) for pre-stress growth, the acute-stress response, and the acclimation phase. (**D**) Cell or two-cell colony size was estimated by the number of pixels within the mask for each colony, and growth rates were calculated based of regression of those points during the pre- or post-stress phases. Cell volume change was reflected in the difference in pixel number before and after stress.

The online version of this article includes the following figure supplement(s) for figure 1:

**Figure supplement 1.** Cellular response to salt within microfluidics device.

**Figure supplement 2.** Growth rate estimates are robust.

separable from the cellular system (defined as factor-specific variation). In contrast, activity-state changes in the upstream signaling networks or cellular system itself could produce coordinated activation of the stress response (referred to as systemic variation). Distinguishing between local versus systemic variation has been difficult, since most studies to date have followed only single transcription factors. We recently developed strains in which two differentially tagged transcription factors regulated by the same signaling network are expressed in the same yeast cell. Msn2 activator fused to mCherry is co-expressed with the transcriptional repressor Dot6 fused to GFP. Both factors help to coordinate the yeast environmental stress response (*Gasch et al., 2000*; *Causton et al., 2001*): whereas Msn2 activates defense genes that are induced in the ESR (iESR genes), Dot6 represses growth-promoting genes involved in ribosome biogenesis that are correspondingly repressed in the ESR during stress (rESR genes) (*Lippman and Broach, 2009*; *Bergenholm et al., 2018*). Both factors are controlled by the Protein Kinase A (PKA) and mTOR pathways, which are generally associated with promoting growth (*Figure 1A*): PKA/TOR-dependent phosphorylation of Msn2 and Dot6 maintains the factors in the cytosol, whereas Msn2 and Dot6 dephosphorylation after PKA/TOR inhibition leads to their nuclear localization (*Görner et al., 1998*; *Smith et al., 1998*; *Lippman and Broach, 2009*). Thus, we expect the two factors to be coordinated in their localization when the stress response is activated systemically but discordant in response to factor-specific differences in regulation.

The challenges in distinguishing factor-specific versus systemic variation have obscured how systemic activation of the stress response relates to other physiological responses. One important factor is growth rate. Growth rate and stress tolerance are competing interests in the cell and are often antagonistically regulated: fast growing cells tend to be the most susceptible to stress and toxins, whereas slow growing or quiescent cells generally survive extreme conditions (*Balaban et al., 2004*; *Lu et al., 2009*; *Zakrzewska et al., 2011*; *Levy et al., 2012*). Part of this antagonistic correlation is thought to be controlled, at least under specific situations, by the RAS-PKA pathway, which promotes growth and suppresses the stress response (*Smith et al., 1998*; *Gasch et al., 2000*; *Zaman et al., 2008*; *Zaman et al., 2009*). *Li et al., 2018* used single-cell microscopy to show that slower growing cells in an isogenic culture displayed lower levels of the PKA allosteric activator cAMP and that artificial activation of PKA diminished the slow growing population (*Li et al., 2018*). They further showed a slight but statistically significant negative correlation between Msn2 nuclear localization and micro-colony growth over the subsequent 10 hr in the absence of stress. This suggests that activation of Msn2 is coupled to reduced growth rate, a theory put forward and debated in other bulk-culture studies (*Regenberg et al., 2006*; *Castrillo et al., 2007*; *Brauer et al., 2008*; *Ho et al., 2018*). The inability to distinguish between factor-specific variation and systemic activation of the stress response likely obscures the true relationship with growth.

Here, we monitored dynamic localization changes of both Msn2 and Dot6 in the same yeast cells, along with a panel of other single-cell measurements, to dissect local and systemic variation and illuminate the relationship between ESR activation and growth rate. We optimized a microfluidics system that can monitor single-cell localization levels and dynamics of both Msn2-mCherry and Dot6-GFP along with single-cell and colony growth rates, size, shape, cell-cycle phase and size changes before and after an acute dose of sodium chloride (NaCl) as a model stressor. Our results revealed several insights, including surprising levels of discordance in Msn2 and Dot6 activation that partly explained variation in post-stress growth rate. We developed a multi-factorial model explaining cell growth rate after stress acclimation to demonstrate that stress acclimation is partly predictable based on prior cellular states. Remarkably, one of the important predictors is the activation level of the Dot6 repressor, which counterintuitively is associated with faster growth acclimation and faster production of stress-induced catalase Ctt1. We discuss implications of this work for understanding how cellular state and transcriptional repression influence stress responses.

## Results

We optimized a microfluidics system that could measure nuclear localization dynamics as well as one- and two-cell colony growth rates before and after exposure to 0.7 M NaCl (*Figure 1B* and Methods). Using this system, we characterized the variation in cell responses for 72 min before and 144 min after exposure to NaCl, which induces ionic and osmotic stress, in biological triplicates done on separate days. This time frame captures phenotypic variation in cells growing in the absence of stress, during the acute stress-response phase (from 0 to 54 min after osmotic stress), and over later timepoints as

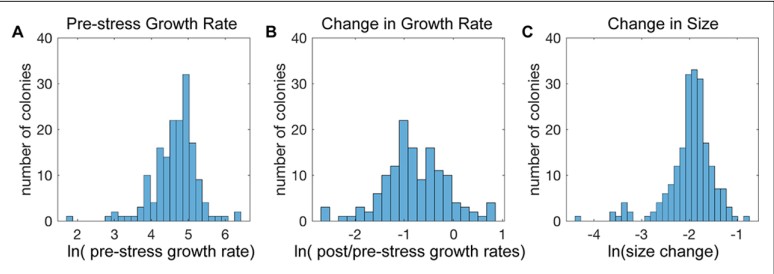

**Figure 2.** Cell-to-cell heterogeneity in the NaCl stress response. (**A-C**) Shown are the distributions of the natural log of (**A**) colony growth rates before stress, (**B**) the change in growth rate after NaCl stress compared to before stress, and (**C**) the maximum change in cell pixel size during the acute-stress response versus during the pre-stress phase.

cells acclimate to continuous NaCl. Microscopy imaging and analysis reports on Msn2-mCherry and Dot6-GFP nuclear localization dynamics in the same cells (*Figure 1C*, *Figure 1—figure supplement 1*). We used MATLAB scripts to identify nuclear translocation events, which we refer to as 'peaks' in the traces (see Methods). We also measured cell and colony growth phenotypes, including colony size, colony growth rates (defined by increase in pixel number of masked colony area and vetted with several analyses, *Figure 1—figure supplement 2*) both before and after stress, and change in cell size due to volume loss upon NaCl stress (*Figure 1D* and Methods). We used the relative change in colony area over time, collapsed from multiple z-stack images per time point, as a proxy for growth rate. One limitation is that growth by this estimation will be under-estimated for cells that bud perpendicular to the slide plane, introducing noise into the growth rate measurements for some cells. We limited our analysis to colonies of only one or two cells at the beginning of the time series and to cells that passed several quality-control filters (see Methods). In total, we analyzed 221 cells passing these filters, collected from the three independent biological replicates.

This system captured variation in all of the features measured. As expected based on previous studies (*Levy et al., 2012*; *Fehrmann et al., 2013*; *Crane et al., 2014*; *Li et al., 2018*; *Jin et al., 2019*), there was substantial variation in cellular growth rates before NaCl addition, confirming that cells vary considerably in their growth properties in the absence of stress (*Figure 2A*). Most colonies

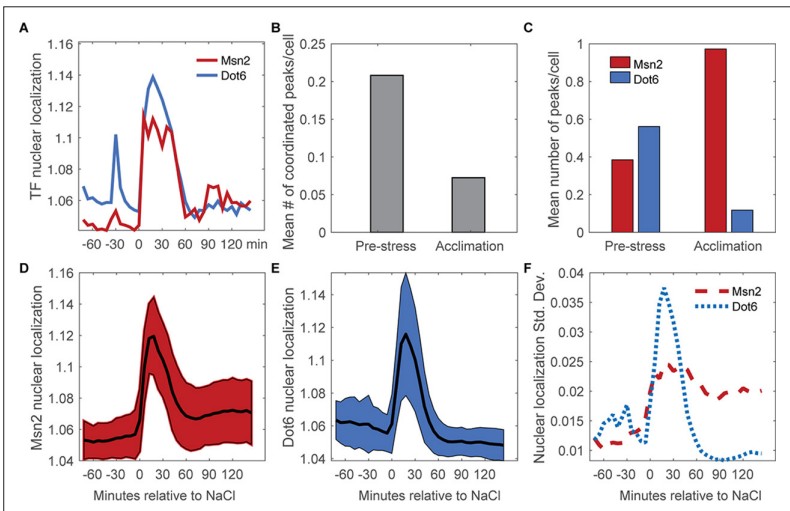

**Figure 3.** Nuclear translocation dynamics of Msn2 and Dot6 are more coordinated before stress. (**A**) Representative traces of Msn2 and Dot6 in the same cell. (**B**) The average number of coordinated peaks for Msn2 and Dot6, *i.e.* peaks called within 6 min (1 timepoint) of each other. (**C**) The average number of nuclear localization peaks per cell for Msn2 (red) and Dot6 (blue) during pre-stress and acclimation phases. (**D–E**) The average (black line)+/- one standard deviation (colored spread) of Msn2 (**D**) and Dot6 (**E**) nuclear localization during the time course. (**F**) Trace of the standard deviation of nuclear localization over the time course for Msn2 (red) and Dot6 (blue).

reduced their growth rate in response to NaCl stress (but not a mock treatment, *Figure 1—figure supplement 2F*), but once again there was substantial variation: some cells showed dramatic growth reduction upon NaCl, whereas others showed little to no change (*Figure 2B*). There were even individual colonies that accelerated growth after stress: 11 of 14 of these cells showed a small bud at the time of salt exposure, suggesting a cell cycle connection. NaCl-induced osmotic pressure is expected to produce rapid water loss before cells acclimate, and indeed most cells shrunk immediately after stress despite substantial variation in size changes (*Figure 2C*). Together, these results highlight the extensive cell-to-cell variation in behavior that is not identified in bulk measures of culture growth.

## Msn2 and Dot6 nuclear localization show only partial coordination

We next investigated co-variation in Msn2-mCherry and Dot6-GFP localization dynamics, before and as cells responded to NaCl. Both factors showed sporadic activation in unstressed cells, with brief and typically low levels of nuclear translocation (*Figure 3A*). Roughly 54% of Msn2 pre-stress peaks and 37% of Dot6 pre-stress peaks were temporally coordinated with the other factor (*Figure 3B*), which is significantly above chance (p<<0.0001, permutation analysis, see Methods) and suggests systemic activation of the stress response. This reveals both coordinated and independent fluctuations in Msn2 and Dot6 activation in the absence of stress, consistent with our prior results (*Gasch et al., 2017*). In the vast majority of cells, NaCl provoked a dramatic and coordinated increase in nuclear localization of both Msn2 and Dot6 (acute phase). However, after stress Msn2 and Dot6 behavior deviated: whereas few cells showed post-stress Dot6 nuclear translocation, many cells showed asynchronous pulses of Msn2 (*Figure 3C–F*), consistent with prior work (*Petrenko et al., 2013*). This was surprising, since we expected that Msn2 and Dot6 would be highly correlated during and immediately after NaCl treatment.

In the course of this analysis, we realized another key difference between Msn2 and Dot6: the profiles of Dot6 nuclear pulses were often highly correlated between unstressed cells in the same colony, indicated by co-occurring peaks in two-cell colonies (*Supplementary file 1*). Permutation tests showed that this was highly significant compared to random chance (p=9.3e-4, see Methods). In contrast, the co-occurrence of Msn2 peaks in cells from the same colony was not significantly different from random. Since these cells are in the same local environment and have a shared life history in that one cell is the daughter of the other, it suggests that some feature of Dot6 regulation is predictable but separable from Msn2 behavior.

## Reproducible differences in Msn2 versus Dot6 activation reveal subpopulations of cells

Comparisons of Msn2 and Dot6 nuclear localization patterns indicated different localization dynamics across cells, raising the possibility of distinct cell subpopulations. To investigate, we used Gaussian finite mixture modelling (*Scrucca et al., 2016*) of the population-normalized Msn2 and Dot6 nuclear localization traces to identify populations or 'clusters' of cells with distinguishable localization patterns (*Figure 4*, see Methods). Most clusters captured cells from all three biological replicates, with the exception of cell Cluster 9 and several small clusters that were enriched for cells from one replicate (*Supplementary file 2*). Six of these patterns were clearly recapitulated in an independent experiment (*Figure 4—figure supplement 1*). Thus, most of the cell groupings represent reproducible subpopulations with different stress-responsive behaviors.

The subpopulations were differentiated by a combination of transcription-factor phenotypes. One distinguishing feature was the level of Dot6 activation during the acute-stress phase. Cluster 11 was characterized by lower than population-median magnitude of acute-stress Dot6 nuclear translocation, whereas cells in Clusters 6 and 7 showed higher-than-median Dot6 response. These results are consistent with the wider variance of Dot6 nuclear translocation levels during the acute phase (*Figure 3D–F*). A second distinguishing feature was the level of nuclear Msn2 and Dot6 before stress. Cluster 11 cells showed low levels of Dot6 before stress, whereas cells in Clusters 9 and 6 displayed higher-than-median nuclear Msn2 and Dot6 during this phase. Finally, the behavior of Msn2 during the post-stress acclimation phase was significantly different across subpopulations. Whereas Clusters 11 and to some extent 7 showed low levels of post-stress Msn2 nuclear localization, cells in multiple clusters showed high levels and/or pulsatile nuclear Msn2 as cells acclimated. We noticed that cells in Clusters 2 and 3 showed elevated levels of mCherry that persisted over time compared to other cells.

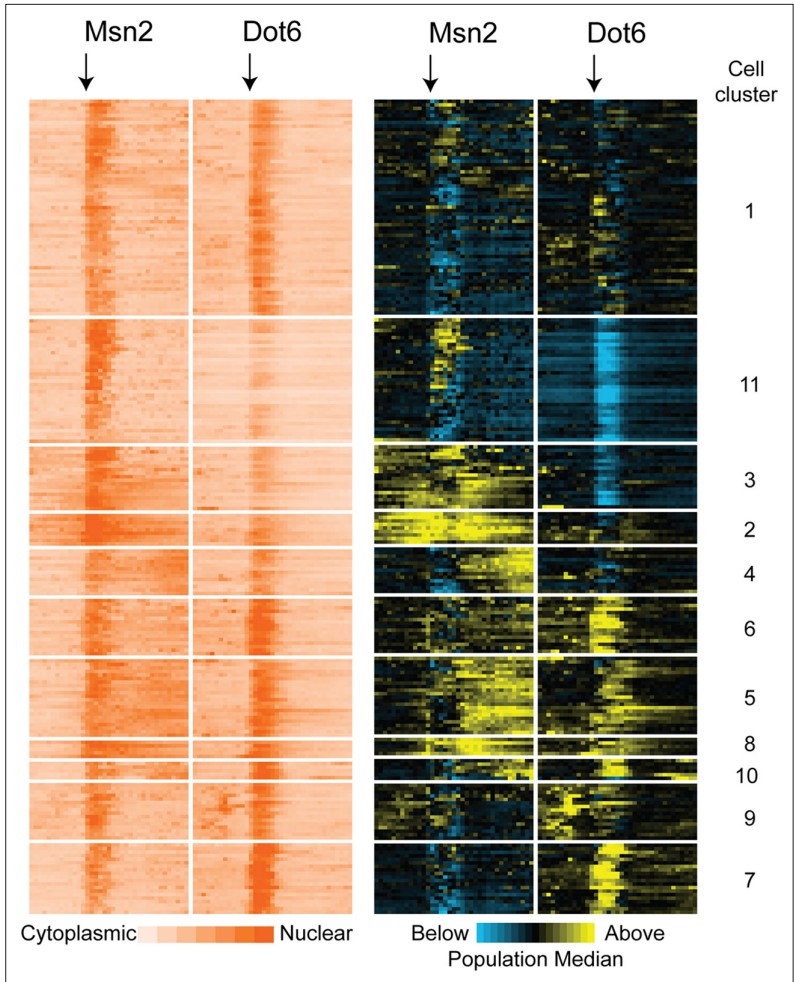

**Figure 4.** Subpopulations of cells show distinct Msn2 and Dot6 translocation dynamics. 221 cells passing quality control metrics were partitioned into sub clusters based on their population-centered nuclear translocation dynamics shown on the right. Each row represents a cell and each column in a block represents a single timepoint; time of NaCl addition is indicated with an arrow. Data on the left show the $\log_2$ ratio of nuclear versus total Msn2 (left) or Dot6 (right) according to the orange-scale key, see Methods. Data on the right show the same data normalized to the population median at each timepoint: yellow values indicate higher-than-median nuclear localization levels and blue indicates lower-than-median nuclear localization. Cell clusters identified by the package mclust are labeled to the right.

The online version of this article includes the following source data and figure supplement(s) for figure 4:

**Source data 1.** The text file contains the $\log_2$ of unnormalized nuclear trace values and the population-median-normalized nuclear trace values across the time course for each cell, divided into clusters (for strain AGY1328).

**Figure supplement 1.** Cellular profiles are recapitulated.

**Figure supplement 1—source data 1.** The text file contains the $\log_2$ of unnormalized nuclear trace values and the population-median-normalized nuclear trace values across the time course for each cell, divided into clusters (for strain AGY1813).

Closer inspection of the microscopy images suggested that some of the signal may not reflect nuclear translocation but instead was likely vacuolar signal (see more below). As mentioned above, the variation in nuclear localization dynamics captured within these clusters occurred in all three biological replicates and in a separate experiment (See *Supplementary file 2* and *Figure 4—figure supplement 1*), indicating reproducible distinctions in transcription factor behavior. Together, this analysis revealed important differences in cellular behavior across the phases of the NaCl response that are obscured by aggregate analysis of all cells in the population.

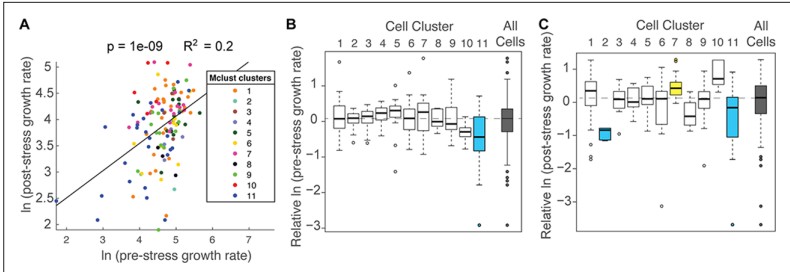

**Figure 5.** Cell subpopulations display different growth rates before and after stress. (**A**) Correlation between the natural log of pre- and post-stress growth rates for each cell, colored according to its cell cluster in **Figure 4**. (**B–C**) Distribution of median-centered growth rates before (**B**) and after (**C**) NaCl addition, for cell clusters shown in **Figure 4**. Boxes are colored yellow or blue if the distribution was significantly higher or lower, respectfully, from all other cells in the analysis (Wilcoxon Rank Sum test, FDR < 0.022). Dashed line indicates the median of all cells analyzed.

The online version of this article includes the following figure supplement(s) for figure 5:

**Figure supplement 1.** The strength of transcription factor nuclear localization is weakly related to cell cycle phase and budding.

**Figure supplement 2.** Cell clusters show similar relationships with pre- and post-stress growth rates.

## Cell subpopulations show different relationships with cell growth

Are subpopulations of cells identified above biologically meaningful? We turned to the other cellular measurements to look for co-variates in cellular behavior that reflect on higher-order relationships (**Figure 5**). We tested each of the cell subpopulations for statistically significant differences in pre-stress growth rate, post-stress growth rate, starting size, volume change, and cell-cycle phase at the time of NaCl exposure (inferred by visual inspection of bud size and nucleus location in the cell, see Methods). We found no significant correlations with cell volume changes or cell-cycle phase (although there was a minor signal for cell cycle, **Figure 5—figure supplement 1**). This is consistent with the lack of strong connection between cell-cycle phase and stress response found in several other studies (**Paek et al., 2016**; **Gasch et al., 2017**; **Bagamery et al., 2020**). In contrast, several clusters showed significant differences in growth rates.

Overall, there was a positive correlation between pre-stress growth rate compared to post-stress growth rate (**Figure 5A**); however, the association was different for subpopulations of cells. Cells in Cluster 11, which were characterized by below-average Dot6 response before and during stress, showed slower growth rates before and after NaCl treatment (**Figure 5B–C**), and the slower growth was consistent when biological replicates were analyzed individually (p<0.02, T-test) and across multiple experiments (**Figure 5—figure supplement 2**). In contrast, cells in Cluster 7 showed higher than average recovery growth rates – these cells were characterized by larger-than-average Dot6 nuclear localization responses and somewhat below-average nuclear translocation of Msn2 during the acute-response phase. The relationships between post-stress growth rate and Dot6 response during the acute phase raised the possibility that this factor's activation is more closely tied to growth rate than Msn2, even when both factors are activated in a systemic response. Interestingly, cells in Cluster 2 that had unusually high (and potentially vacuolar) mCherry fluorescence before stress displayed very slow growth recovery after stress, demonstrating the biological validity of the subpopulation and raising the possibility of poor stress acclimation in these cells. (We note that cells with apparent vacuolar signal were excluded from subsequent analyses).

## Combining multiple characteristics increases the predictive power to explain post-stress growth rate

The above results hinted that how well cells acclimate to NaCl stress, as indicated by post-stress growth rate, may be predicted by cellular responses both before and during the stress response. Based on the work of **Li et al., 2018**, we expected a negative correlation between Msn2 nuclear localization and growth rate (which they reported over much longer time frames). While there was no correlation with pre-stress growth rate (p=0.65), we did observe a negative correlation between

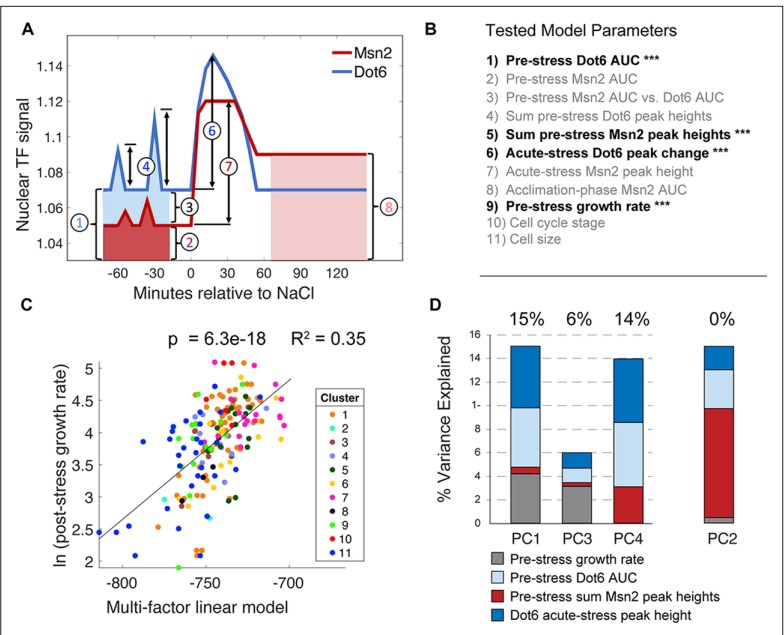

**Figure 6.** A multi-factor model best explains variation in post-stress growth rate. (**A**) A representation of the nuclear localization measurements used in the multi-factor linear regression model. (**B**) Factors considered in the multi-factor linear regression model; those with significant contributions are highlighted with ***. (**C**) The variance in ln(post-stress growth rate) explained by the multi-factor linear regression model. P-value and $R^2$ are shown at the top of the plot and cell subcluster is indicated according to the key, showing that no single cluster dominates the correlation. (**D**) Principal component (PC) regression of post-stress growth rate and deconvolution of contributing factors according to the key. Variance explained is listed at the top of each bar (where PC2 does not contribute to post-stress growth rate).

The online version of this article includes the following figure supplement(s) for figure 6:

**Figure supplement 1.** Linear regression of individual parameters on post-stress growth rate.

**Figure supplement 2.** Dot6 acute-stress peak height correlates with post-stress growth rate even across cells with no difference in pre-stress growth.

pre-stress Msn2 activation (taken as the area under the nuclear-localization curve (AUC) for pre-stress timepoints) and post-stress growth rate; however, the correlation explained only 3% of the variance (p=0.016, linear regression), indicating that the pre-stress behavior of Msn2 has little power to predict post-stress growth rate in our study.

We next investigated other features that could explain differences in post-stress growth rate (*Figure 6A–B*). Pairwise correlations revealed that some individual features, such as the magnitude of Dot6 acute-stress response, correlated well with post-stress growth rate but others did not (*Figure 6—figure supplement 1*). However, the most impactful single factor – pre-stress growth rate – explained only 20% of the variance in post-stress growth rate (*Supplementary file 3*).

We next asked if combining cellular phenotypes into a single multiple linear model could explain more of the variance in growth. We considered multiple metrics for summarizing pre-stress nuclear localization, including AUC (which is a measure of the overall nuclear abundance) and the sum of called translocation peak heights (which is influenced by the magnitude and frequency of pre-stress pulses), along with acute-stress translocation peak height and AUC during the acclimation phase. The model also incorporated other cell features including pre-stress growth rate, cell-cycle phase at the time of NaCl exposure, and cell size factors (See 'Model 1' in *Supplementary file 3* for all parameters used). Factors that did not contribute significantly (adjusted p>0.05) were progressively removed until the variance explain decreased (*Supplementary file 3*). The final regression identified four factors that contributed significantly to explain post-stress rate ('Model 3' in *Supplementary file 3*): pre-stress AUC of Dot6 nuclear localization, the sum of pre-stress Msn2 peak heights, the pre-stress growth rate of the cells, and the magnitude of Dot6 nuclear localization change immediately after NaCl. Together, these factors – all but one of which represent pre-stress cellular phenotypes – explained 35% of the

variance in post-stress growth rate (*Figure 6C*), nearly doubling the explanatory power of any single feature alone. We note that noise in the growth-rate measurements is likely diminishing the true fit, such that the explanatory power reported here is actually an under-estimate.

One challenge is that several of these phenotypes could be co-variants of an underlying hidden variable or cellular state. For example, both pre-stress growth rate and Dot6 acute-stress peak height correlate with post-stress growth rate, but they also correlate with each other: cells growing faster before stress have a larger Dot6 stress-response. The mixed-linear model reports that both factors contribute separable predictive power, and indeed together they explain more of the variance in stress acclimation than either factor alone. Nonetheless, to further disentangle their co-variation, we applied principal component (PC) regression. We first analyzed the four statistically-significant model-input variables in *Figure 6B* by PCA and then used the resulting components as factors in a linear model of post-stress growth rates (see Methods). PC1 and PC3 together explained 21% of the variance in post-stress growth rate: both captured co-variation in pre-stress growth rate, acute-stress Dot6 response, and pre-stress transcription factor behaviors, indicating that these features likely reflect the same aspects of the cellular state (*Figure 6D*). However, PC4 that is dominated by Dot6 behavior but not influenced by pre-stress growth rate explained an additional 14% of growth acclimation (p=1e-4). A fourth component, PC2, was dominated by pre-stress Msn2 behavior but showed no power to predict post-stress growth acclimation rates. Thus, behavior of the Dot6 repressor independently correlates with post-stress growth rate. As further confirmation, we analyzed the correlation between Dot6 acute-stress peak height and post-stress acclimation in a subset of cells with similar pre-stress growth rates. Indeed, pre-stress growth rate had no predictive power for this subset of cells, whereas Dot6 peak height explained 12% of the variance (p=1e-4, *Figure 6—figure supplement 2*). Thus, the behavior of the Dot6 repressor during acute NaCl stress is associated with growth recovery as cells acclimate (see Discussion).

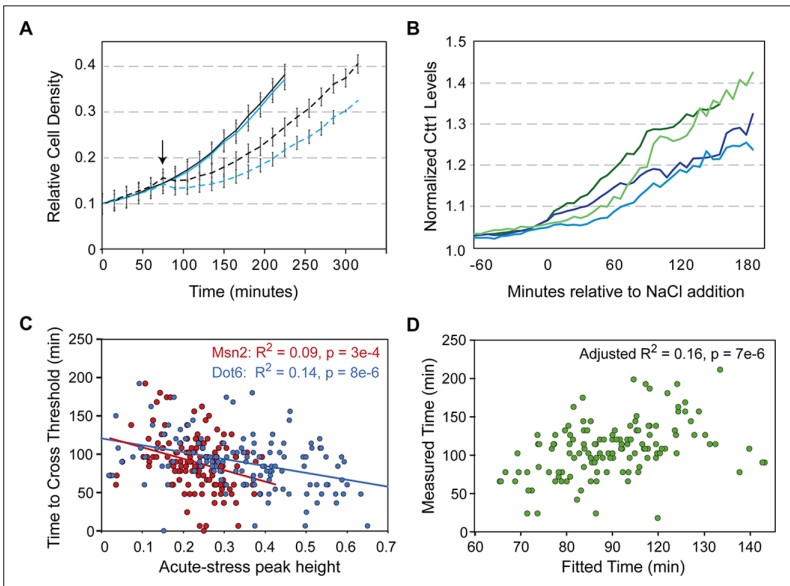

**Figure 7.** Dot6 activation correlates with faster Ctt1 production. (**A**) The average and standard deviation (n=4) of growth rates of wild-type (black lines) and *dot6Δtod6Δ* cells (blue lines) in the absence (solid) and presence (dashed) of 0.7 M NaCl added at 75 min (arrow). (**B**) Representative traces of single-cell Ctt1 production for pairs of cells that reach similar levels of Ctt1. (**C**) Correlation of Ctt1 production timing (time to change 5%) versus acute-stress peak heights. (**D**) The two-factor model correlates with measured Ctt1 production time, with only marginal contribution of Msn2 peak height (p=0.053). Adjusted $R^2$ is shown in both figures.

The online version of this article includes the following figure supplement(s) for figure 7:

**Figure supplement 1.** Dot6 acute-stress response is correlated with pre-stress transcription factor behaviors.

## Dot6 activation is associated with faster production of Ctt1 protein

Dot6 is the transcriptional repressor of growth-promoting ribosome biogenesis (RiBi) genes; thus, its positive association with post-stress growth rate may seem counterintuitive. However, this result is consistent with past work from our lab: in response to NaCl stress, cells lacking *DOT6* and its paralog *TOD6* fail to repress hundreds of genes in the RiBi regulon (*Lee et al., 2011*; *Ho et al., 2018*). These transcripts remain associated with ribosomes, whereas stress-induced transcripts including Msn2-regulated *CTT1* show reduced ribosome association (*Ho et al., 2018*). Despite producing more *CTT1* mRNA, the *dot6Δtod6Δ* mutant shows delayed production of Ctt1 protein. We proposed that transcriptional repression of otherwise highly transcribed mRNAs is important to free up translational capacity to translate stress-induced transcripts (*Ho et al., 2018*).

To investigate on a cellular level, we attempted microscopy in a *dot6Δtod6Δ* strain; however, whereas the strain grew fine in the device before stress, it was unable to recover growth after NaCl treatment. Indeed, bulk-culture experiments revealed that the *dot6Δtod6Δ* mutant grew as wild type before stress, but showed significantly reduced growth rate after NaCl treatment (*Figure 7A*). This is consistent with our results in wild-type cells, where cells with a weaker Dot6 response show a reduced post-stress growth rate. Thus, bulk-culture experiments reinforce the trends of the microfluidic analysis, showing that Dot6 provides a protective response during NaCl stress.

A major unanswered question is how Dot6 behavior in a *wild-type* cell relates to growth and Ctt1 production. We therefore generated a strain to track Dot6-GFP, Msn2-mCherry, and Ctt1-iRFP in the same cells. Cellular Ctt1 levels (defined as maximum iRFP signal normalized to pre-stress levels, see Methods) were correlated with both Msn2 and Dot6 peak heights (but not their pre- or post-stress behaviors). However, the explanatory power was significantly higher when considering the timing of Ctt1 production. We defined the time for Ctt1-iRFP levels to cross a change threshold (see Methods). Even for cells that reached the same maximal Ctt1 levels, the time to get there varied (*Figure 7B*). We found that the time to cross that threshold was correlated with both Msn2 and Dot6 peak heights, which are themselves weakly correlated; however, the variance explained was significantly higher for Dot6 activity (*Figure 7C*). Indeed, a mixed model considering both factors confirmed that the contribution of Dot6 was significantly more than that of Msn2 behavior, which was only marginally significant in the model (p=0.053, *Figure 7D*). Dot6 is not known to regulate Ctt1 or bind its promoter (*Zhu et al., 2009*), and we previously showed that *dot6Δtod6Δ* cells induce *CTT1* transcript to higher levels than wild type during NaCl stress (*Ho et al., 2018*). Together, this suggests an indirect effect of Dot6 that is separable from Msn2 regulation. In sum, our results indicate that Dot6 provides a protective response during NaCl treatment (*Figure 7A*), is correlated with faster Ctt1 production in both mutant (*Ho et al., 2018*) and wild-type cells (*Figure 7D*), and is associated with faster growth recovery after NaCl treatment (*Figure 6*, see Discussion).

## Discussion

By following dynamic activation of two different stress-regulated transcription factors, in conjunction with other cellular features including growth rate, cell size, and cell cycle stage, we uncovered previously unrecognized inter-dependencies that present new insights into mechanisms of stress defense. Our results reveal much more complexity in Msn2 and Dot6 behavior than previously recognized, that the relative activation of these factors along with other pre-stress phenotypes can partly predict cellular outcomes including growth acclimation, and that behavior of the Dot6 repressor influences post-stress growth rate and the dynamics of a downstream response. Below we discuss implications of these results.

## Complexities in Msn2 dynamics reflect diversity in stress-responsive states

Past studies focusing on aggregate analysis of all single cells in the population reported condition-specific dynamical behavior of Msn2, such as prolonged nuclear pulsing after glucose starvation versus a burst of activation before acclimating to osmotic stress (*Hao and O'Shea, 2012*; *Petrenko et al., 2013*; *AkhavanAghdam et al., 2016*). Elegant studies by Hansen et al. used artificial activation of Msn2 (through chemical inhibition of PKA activity) to show that these differences in Msn2 nuclear translocation dynamics produce different transcriptional outputs (*Hansen and O'Shea, 2013*;

*Hansen and O'Shea, 2015a*; *Hansen and O'Shea, 2015b*; *Hansen and O'Shea, 2016*). Target-gene promoters display different dependencies on the amplitude, frequency, and duration of Msn2 nuclear translocation, such that distinctions in Msn2 behavior activate different sets of genes. Comparable studies of regulators in mammalian systems also reported stress-specific differences in the dynamics of nuclear translocation, which correspond to differences in gene activation (*Purvis et al., 2012*; *Kracikova et al., 2013*; *Paek et al., 2016*). One limitation of the approach of Hansen et al. is that activating Msn2 by wholesale inhibition of PKA likely loses much of the heterogeneity seen in natural responses. Our study thus provides an important complement to artificial system activation.

In fact, our analysis revealed highly varied responses across subpopulations of cells responding to the same stress stimulus. Some cells responded to the osmotic/ionic stress induced by NaCl with a large nuclear pulse of Msn2 followed by near complete acclimation, as previously reported for sorbitol-induced osmotic stress – but other cells showed extensive and prolonged Msn2 fluctuations during the acclimation phase, akin to what has been reported for glucose starvation. These subpopulations are obscured by aggregate analysis but have important implications, since the different dynamics of Msn2 (and likely also Dot6) activation produce different transcriptomic outputs, even for cells responding to the same stressor in the same environment. This hypothesis is consistent with past work from our lab investigating single-cell transcriptomics, in which isogenic cells in the same culture displayed different transcriptomes upon NaCl stress, including for ESR genes, indicating that they experience the stress differently (*Gasch et al., 2017*).

## The Dot6 repressor provides a protective response during stress

Although the variety in Msn2 responses likely has important consequences on downstream gene expression, we were surprised to find little connection to growth rate, at least in the short time frames studied here. Instead, the response of Dot6 explained a much larger fraction of the variance in post-stress growth rate, when considered alone or in the multi-factor linear model (*Figure 6* and *Supplementary file 3*). Cells with a larger Dot6 response during the acute-stress phase showed faster production of Ctt1, separable from Msn2 activity (*Figure 7*), and faster growth recovery during the acclimation phase. In contrast, cells completely lacking Dot6 and its paralog show delayed Ctt1 accumulation despite having more transcript (*Ho et al., 2018*) and dramatically reduced post-stress acclimation (*Figure 7A*).

These results are consistent with our working model of Dot6 activity. At least in response to NaCl treatment, transcriptional repression does not lead to reduced abundance of the encoded proteins (*Lee et al., 2011*). Instead, we proposed that transcriptional repression helps to deplete the pool of RiBi transcripts that are normally highly transcribed and translated in actively growing cells (*Lee et al., 2011*; *Ho et al., 2018*). In the absence of Dot6 repression, aberrantly abundant RiBi transcripts compete with induced mRNAs for available translational machinery, thereby delaying translation of stress-defense transcripts. In the case of NaCl, the limiting factor is unlikely to be ribosomes: we previously showed that this yeast strain exposed to the same dose of NaCl removes a population of ribosomes from the translating pool immediately after stress (*Ho et al., 2018*). This is consistent with bacterial models of growth regulation, in which cells preserve some ribosomes for later stress acclimation (also indicating that growth rate under these conditions is not limited by ribosome availability) (*Mori et al., 2017*; *Kim et al., 2018*; *Korem Kohanim et al., 2018*; *Remigi et al., 2019*; *Wu et al., 2022*). Evidence from bacteria and incidental results in yeast suggest that other features related to translation elongation may limit cell growth in this situation (*Dai et al., 2018*; *Ho et al., 2018*; *Wu et al., 2022*), a limitation that may be alleviated by removing some ribosomes from the translating pool. How all of this fits into broader cellular states is discussed below.

## Differences in pre-stress cellular states influence stress acclimation

Many studies have found significant variation in how cells respond to acute stress. Using our system and the conditions studied here, upwards of 35% of the variance in post-stress growth rate could be explained by a multi-factorial model that includes both pre-stress and acute-stress phenotypes. It will be interesting to see as technology develops for improved growth rate measurements if this fit improves further. The remaining unexplained variation is likely influenced by additional features of the cellular state, as well as stochastic effects. We found no connection to cell-cycle phase or cell size, although the lack of correlation could be masked by other confounders (*Barber et al., 2021*). But

one likely contributor is differences in pre-stress metabolic or mitochondrial states as implicated in several studies (*Fehrmann et al., 2013*; *Gasch et al., 2017*; *Laporte et al., 2018*; *Dhar et al., 2019*; *Bagamery et al., 2020*). Bagamery et al. showed that pre-stress fluctuations in fermentative versus respirative metabolism influence how cells recover from glucose starvation, with antagonistic fitness effects depending on the situation (*Bagamery et al., 2020*). Other studies linked variation in mitochondrial function and morphology to cell age and the ability to enter quiescence, which could also influence stress responsiveness (*Fehrmann et al., 2013*; *Laporte et al., 2018*). An interesting avenue for future investigation would be to measure metabolic and mitochondrial states along with features studied here.

Regardless, our results are consistent with the fact that pre-stress cellular states influence how cells will respond to future stress. Some cells in our study were fast growing before stress, showed a larger Dot6 response during stress, and acclimated faster in terms of post-stress growth rate; in turn, cells that were slow growing before stress had lower pre-stress Dot6 activity, lower Dot6 activation during the acute phase, and a slower growth acclimation. One hypothesis is fast-growing cells may have higher biosynthetic capacity, and thus more need for ribosomes and higher transcription of RiBi genes. These cells may therefore need to slam on the brakes of RiBi production more strongly in order to free up translational capacity. Repression of RiBi transcripts in and of itself need not impact subsequent growth recovery, if cells already harbor ample ribosomes at the time of stress.

On the other hand, the size of the Dot6 acute-stress peak correlates with post-stress growth acclimation in a way that can be separated from pre-stress growth rate (*Figure 6C, D and – Figure 6—figure supplement 2*). Thus, some cells may be growing at average rates but still require a large Dot6 response, for example if they are already somewhat limited in translational capacity for other reasons and therefore require a strong Dot6 response. Interestingly, pre-stress growth rate did not correlate with the time to cross the Ctt1 threshold (p=0.24), indicating that the correlation with Dot6 is independent. Future studies will be required to test these hypotheses. Interestingly, the Dot6 acute-stress peak height can be fairly well predicted by the *relative* pre-stress activity of Msn2 versus Dot6 ($R^2$=0.42, *Figure 7—figure supplement 1*), again linking acute-stress behavior to pre-stress cell states.

Our work adds to a growing body investigating the relationship between stress defense and growth rate. While we expected a relationship between coordinated Msn2/Dot6 activation and growth rate based on past studies (*Brauer et al., 2008*; *Ho et al., 2018*), we instead discovered unexpected discordance in the factors' behavior and an unexpected association of acclimation growth rate and Dot6 activity, the opposite of what several past models predict (*Regenberg et al., 2006*; *Castrillo et al., 2007*; *Brauer et al., 2008*; *Airoldi et al., 2009*). These results highlight the complexities of eukaryotic growth control and set the stage for further dissection of the driving regulators of growth rate and how best to predict growth under fluctuating conditions.

## Methods

Strains used include AGY1328 (BY4741 *DOT6-GFP*(S65T)-His3MX, *MSN2-mCherry-HYGMX*), AGY1813 (BY47141 *DOT6-GFP*(S65T)-His3MX, *MSN2-mCherry-HYGMX, CTT1-iRFP-KanMX*), and AGY1363 (BY4741 *dot6::KAN tod6::HYG CTT1-GFP(S65T)-His3MX*) (strains available upon request). For microscopy experiments, overnight cultures were grown from single colonies to exponential phase at 30°C (Optical Density, $OD_{600}$ <1) in Low Fluorescent Medium (LFM) before cells were adhered to the microscope slide as described below. LFM consisted of 0.17% Yeast Nitrogen Base without Ammonium Sulfate, Folic Acid, or Riboflavin (#MP114030512, Thermo Fisher Scientific, Waltham, Massachusetts), 0.5% Ammonium Sulfate, 0.2% complete amino acids supplement, where individual amino acids concentrations are as defined in Yeast Synthetic Drop-out Media Supplements (Sigma-Aldrich, Saint Louis, Missouri), and 2% Glucose. Cells were grown in LFM shake flasks at 30°C for data shown in *Figure 7A*.

An FCS2 chamber (Bioptechs Inc, Butler, Pennsylvania) microfluidic system was used for time-lapse microscopy. In short, a 40 mm round glass coverslip and FCS2 lower gasket were assembled, and Concanavalin A solution (2 mg/mL Concanavalin A, 5 mM $MnCl_2$, 5 mM $CaCl_2$) was applied to the coverslip, incubated for 2 min, then aspirated. Next, 350 µL of an ~0.5 $OD_{600}$ culture was placed on the coverslip and incubated 5 min for cells to settle and adhere to the Concanavalin A. 150 µL of the media was then removed and the rest of the FCS2 chamber was assembled.

Media was flown through the FCS2 chamber using gravity flow. Input tubing was attached to elevated bottles containing either LFM or LFM +0.7 M NaCl (See diagram in *Figure 1B*) with a valve to switch between media with and without 0.7 M NaCl. The outflow tubing was connected to an additional ~1 m of BD Intramedic PE Tubing (#1417012D, Thermo Fisher Scientific, Waltham, Massachusetts) with the smaller inner diameter of 0.86 mm being vital to controlling the gravity flow of media. The entire assembly, including the microscope stand, bottles containing media, and FCS2 chamber, were enclosed in an incubator maintaining internal temperature of 30°C throughout the entire protocol.

A Nikon Eclipse Ti inverted microscope with the Perfect Focus System (Nikon Inc, Melville, New York) was used for time-lapse microscopy. The GFP signal was captured using a ET-EGFP single band filter cube (#49002, Chroma Technology Corp, Bellows Falls, Vermont excitation 470/40 x emission 525/50 m). The mCherry signal was captured using a ET/mCH/TR single band filter cube (#96365, Chroma Technology Corp, Bellows Falls, Vermont excitation 560/40 x emission 630/75 m). In addition, exposure from a halogen lamp was used to capture white-light images of all cells. For experiments using AGY1813, the iRFP signal was captured using a Cy 5.5 filter cube (#49022, Chroma Technology Corp, Bellows Falls, Vermont excitation 650/45 x emission 720/60 m).

Images of each field of view were captured at 6-min intervals. The z-focal plane focus was set on the center of cells, and images were taken 1 μm above, at, and 1 μm below this center of focus, generating a three-image z-stack for each channel. The three-image z-stacks were collapsed into a single image by taking the maximum projection of the 3 images using a custom MATLAB script.

Cells were identified using a MATLAB circle-finding function on the brightfield images. Individual cells were then tracked through all images using the MATLAB *simpletracker* function (*Tinevez, 2019*). Cell colonies were defined by segmenting images into a binary black-and-white image, and single colonies were defined as enclosed masks. The number of cells within each colony was determined simply as the number of identified circles that overlapped with a given enclosed white area of the binary images. Pre-stress growth was scored by linear regression on colony size (defined as the total pixel number within the masked area of the colony) for the first twelve 6-min time points and reported as the natural log of the rate of increase. Post-stress growth was measured in the same manner for time points 20–29 (representing resumed growth at the beginning of the acclimation phase: 114–168 min into the time course). We note that our proxy for post-stress growth rate, taken as an indication of how well cells acclimate to salt stress, could also be influenced by differences in volume recovery for some cells, which may also be a feature of successful acclimation.

We applied several quality control filters to insure accuracy of growth rates. First, to ensure that colony growth rates were representative of nuclear localization dynamics within individual cells, we limited our analysis to colonies consisting of no more than two cells at the time points leading up to NaCl exposure. Most of these two-cell colonies represented mother/daughter cells and therefore had clear shared life histories. Second, in some cases a budding daughter cell was lost during the time-course, resulting in a misleading negative growth rate. Consequently, regressions resulting in negative slopes were excluded. Lastly, a visual inspection of individual colonies during the time course excluded colonies where new cells adhered to a given colony. Thirty cells were excluded from post-stress measures due to these cell adhesion issues that skewed colony size measures. Another six cells (2.7% of total cells) had no apparent post-stress growth and the calculated slope was therefore dominated by noise in pixel number. This resulted in either a negative or near zero slope and consequently did not provide an informative growth rate measure when taking the natural log of the change in colony size. Consequently, these six cells were also excluded from post-stress growth rate measures. Experiments with AGY1813 (n=3) had the same quality control filters applied to them, with an additional metric applied to exclude cells expressing persistent, high iRFP signal throughout the time course (11 cells). This resulted in an analysis of 228 cells.

Cell-cycle phase at the time of osmotic stress was measured by visual inspection of cell bud presence/size and nucleus location within the cell in accordance with standard yeast cell-cycle definitions (*Howell and Lew, 2012*). Specifically, S-phase appearance of a bud but no migration of nucleus, (G2) bud and nucleus migration toward bud, but no nucleus in daughter cell, M-phase nucleus in both cell and bud, and active division of nuclei, (G1) no bud and nucleus is not actively dividing.

Nuclear localization of Msn2 and Dot6 was measured by taking the pixel intensity of the top 5% of pixels in the cell divided by the median pixel intensity within the circle mask identified for each cell,

similar to other studies (*Cai et al., 2008*; *Hao and O'Shea, 2012*; *Petrenko et al., 2013*; *Lin et al., 2015*; *AkhavanAghdam et al., 2016*; *Gasch et al., 2017*; *Granados et al., 2018*). The following nuclear localization metrics were analyzed:

### Nuclear localization peaks

Temporal peaks of nuclear localization were identified using the MATLAB *findpeaks* function, where a peak height is called from a local maximum to the nearest minimum ('valley') on either side of the peak. In order to estimate a threshold for a true peak of nuclear localization versus background noise, a linear regression was done on pre-stress nuclear localization time points to calculate the difference of each point from the regression line, resulting in a baseline standard deviation of localization values. Since from visual inspection of traces and cells there were many more true peaks for Dot6, the standard deviation for the Msn2-mCherry channel was used to calculate this baseline threshold for both Msn2 and Dot6. Specifically, two standard deviations from the mean of the distribution of was used as a threshold. This threshold appeared to be accurate by visual inspection of cells, where the threshold distinguished what looked like true nuclear localization from the images.

### Area under the curve (AUC) of nuclear localization

For pre-stress time points, AUC was calculated by summing the first 9 measurements of nuclear localization scores (top brightest 5% of pixels over the median cellular signal). This summation represents the total relative levels of nuclear localization between all cells. The same AUC calculation was done for the acclimation phase using time points 24–37. The difference in AUC between the two signals (Msn2 – Dot6 AUC in *Figures 6 and 7*) is simply the difference of the two individual AUC measurements.

### Acute stress peak height

The acute stress peak height was calculated by taking the maximum of nuclear localization score during the acute stress response (time points 13–20) and then subtracting the minimum of the nuclear localization scores just before stress (time points 11–13).

iRFP fluorescence was recorded as the median pixel intensity within cell masks, divided by the background fluorescence measured for each image using ImageJ (*Abràmoff et al., 2004*). Maximum Ctt1 levels were taken as the maximum fluorescence signal from T12-T43 timepoints minus the median of pre-stress (T1-T11) signal. Threshold analysis was done by identifying the time it took each cell to cross a 5% change in Ctt1 abundance. Cells that did not cross that threshold were not included in the timing analysis (but were included in correlations with maximum Ctt1 production).

### Cell clustering to identify subpopulations

Nuclear localization scores were $\log_2$ transformed, and for each cell and each factor, the value at each timepoint was normalized to the median of all cells for that factor and time point (*Figure 4*, blue/yellow scale data). The population-median-normalized vector for Msn2 and Dot6 were concatenated and clustered by mclust (*Scrucca et al., 2016*) using model EII and k=30 (which was collapsed to k=11 by mclust for data shown in *Figure 4* and k=9 for data shown in *Figure 4—figure supplement 1*). The $\log_2$ of unnormalized nuclear traces for each cell was added for display in *Figure 4* and supplement (orange/white scale data). Relationships with logged growth rate data before and after stress, calculated as described above, were scored for each cluster of cells compared to all other cells in the data (*Figure 5* and supplements, Wilcoxon Rank Sum test).

Visual inspection of cells within mclust clusters 2 and 3 indicated that some Msn2 signal was focused but outside the nucleus (evidence after NaCl treatment), likely in the vacuole. There were 18 cells were this was observed visually. Since the impact of this signal was uncertain, these cells were excluded from subsequent regression modeling (i.e. *Figures 5A–7*).

### Probabilities of the number cells from each of the three biological replicates

Binomial probabilities were used to determine if each cluster contained more cells from one of the three biological replicates than would be expected by chance. Specifically, if $x$ is the number of cells from a given biological replicate present in a cluster, then probability of having $x$ cells or more in the cluster is

$$\Pr\left(X \ \ x\right) = \sum_{k=X}^{n} \binom{n}{k} p^k \left(1-p\right)^{n-k}$$

where $n$ is the number of cells in the cluster and $p$ is the expected probability of having a cell from a given replicates (that is, the total number of cells in the replicate divided by the total number of cells in all three replicates). Since clusters 8 and 10 had a total of 5 and 6 cells, respectively, they lacked statistical power and were excluded from the analysis.

The Holm-Bonferroni method was used for multiple hypotheses correction, where there were n=27 (9 clusters and testing the number of cells from 3 biological replicates in each cluster) and = 0.05. Using this threshold, only Cluster 9 showed strong enrichment for cells from one replicate compared to expected after multiple-test correction. Of note, this was the only cluster (besides Clusters 8 and 10), that had zero cells from a given replicate (*Supplementary file 3*).

### Permutations of nuclear localization peak matches

To identify if matched peaks of Msn2 and Dot6 were more coordinated than expected by chance, permutations were performed where a random Msn2 and Dot6 trace, including the time points of the called peaks, were randomly paired from the entire dataset. Coordinated peaks were then calculated from these random Msn2/Dot6 trace pairs. These permutations indicated that the number of matched Msn2/Dot6 peaks per cell was much higher than expected by random combinations (zero permuted datasets out of 100,000 total had 0.21 matched pre-stress peaks per cell or more). The same test was done for the matched peaks during the acclimation phase, and although the number of matched peaks per cell was significant for the acclimation time points (a fraction of $4 \times 10^{-4}$ of permuted datasets had 0.07 matched peaks per cell or more), this was significantly less than that for the pre-stress time points. This again demonstrated that there was more coordination in nuclear localization between Msn2 and Dot6 during the before stress compared to after stress.

### Permutations of nuclear localization peak correlations between cells in two-cell colonies

There were 56 two-cell colonies in the dataset. Of these, 15 colonies showed coordinated Dot6 peaks between the two cells, defined as peaks, occurring within one time point of each other. Permutations were performed where the 112 cells from these 56 colonies were randomly assigned in pairs and the same coordinated peak measurements were performed. Similarly, permutations were performed on matched peaks of Msn2. Results are shown in *Supplementary file 1*.

### Linear models

Multiple linear regressions shown in *Figure 6* and *Supplementary file 3* were performed using *fitlm* in MATLAB. Each model was represented by.

$$y = \beta_0 + \beta_1 x_1 + \beta_2 x_2 + \ldots + \beta_n x_n + \epsilon$$

where the dependent variable $y$ is the post-stress growth rate, $\beta_0$ is the intercept, each subsequent $\beta$ is the estimate of the slope for each independent variable $x$, and is the error term. A list of independent variables is shown in *Supplementary file 3* for each of the multiple linear regression performed. The p-values shown in *Supplementary file 3* were determined from the t-statistic of each $\beta$ coefficient was not equal to zero. In *Supplementary file 3*, Model 1 included all variables in the model. Model 2 only included the significant independent variables from Model 1. Model 3 excluded Msn2 acclimation AUC and cell/colony size from Model 2 as the p-values did not pass Holm-Bonferroni correction (=0.05 and $n = 14$. Since Model 3 gave the four most-significant variables, Model 4 then removed pre-stress growth rate to see the resulting explained variance. Model 5 measured the explained variance of the two most significant variables: Dot6 acute stress peak height and pre-stress growth rate. Trends and significance were the same when analyzing only single-cell colonies, except that the minor contribution of the sum of prestress Msn2 peaks to the original model was no longer significant.

For principle component regression (*Figure 6D*), principle component analysis (PCA) was performed using *pca* in MATLAB on the 4 factors that had significant influence on post-stress growth rate (*Figure 6B*, bold). The resulting PCA coefficients (i.e. the loadings) represent the contribution of each of these 4 factors to each PC. For each PC, the value of each coefficient was divided by the sum of coefficient values to give a fractional contribution of each factor to each PC (*Drummond et al.,*

*2006*). A linear model was then performed as described above, where the dependent variable, y, was again the post-stress growth rate, but the independent variables, x, were the resulting PCA scores for each of the 4 factors.

## Acknowledgements

We thank Stephanie Geller, Taylor Scott, and Kieran Sweeney for help on microfluidics and microscopy, Michael Newton and Kirin Hong for statistical discussions, and members of the Gasch Lab for constructive comments. This work was supported by NSF grant 1715324 to APG and NIH 1R35GM128873 to MNM, who holds a Career Award at the Scientific Interface from the Burroughs Welcome Fund.

## Additional information

### Funding

| Funder | Grant reference number | Author |
| --- | --- | --- |
| National Science Foundation | 1715324 | Audrey P Gasch |
| Burroughs Wellcome Fund | 1R35GM128873 | Megan N McClean |
| National Institute of Health and Medical Research | R01CA229532 | Audrey P Gasch |

The funders had no role in study design, data collection and interpretation, or the decision to submit the work for publication.

### Author contributions

Andrew C Bergen, Conceptualization, Data curation, Software, Formal analysis, Validation, Investigation, Visualization, Writing – original draft, Writing – review and editing; Rachel A Kocik, Data curation, Software, Formal analysis, Validation, Investigation, Visualization, Writing – review and editing; James Hose, Resources; Megan N McClean, Resources, Methodology, Writing – review and editing; Audrey P Gasch, Conceptualization, Formal analysis, Supervision, Funding acquisition, Visualization, Writing – original draft, Project administration, Writing – review and editing

### Author ORCIDs

Andrew C Bergen http://orcid.org/0000-0003-1295-7718
Rachel A Kocik http://orcid.org/0000-0003-2422-4538
Audrey P Gasch http://orcid.org/0000-0002-8182-257X

### Decision letter and Author response

Decision letter https://doi.org/10.7554/eLife.82017.sa1
Author response https://doi.org/10.7554/eLife.82017.sa2

## Additional files

### Supplementary files

• Supplementary file 1. Permutations of coordinately timed peaks in cells in two-cell colonies.

• Supplementary file 2. Cell subpopulations are identified in multiple biological replicates. The number of cells in each mclust cluster from *Figure 4* is shown along with the number of those cells from each of three biological replicates. P-values from binomial probability tests (see Methods) are shown and those significant after Holm-Bonferroni correction (namely Cluster 9 which was enriched for cells from replicate 3) are indicated with an asterisk.

• Supplementary file 3. Multiple Linear Models: variables, significance, and explained variance.

• MDAR checklist

• Source code 1. Cell Identification Script. This MATLAB script takes halogen images and identifies

cells. Makes an output file to be used by the 'Source-Code-2.m' script.

• Source code 2. Cell Tracking Script. This script takes the output file from Source-Code-1.m and tracks individual cells through time series and extracts cell size and TF nuclear localization.

• Source data 1. Source data 1 for AGY1328 Single Cell Measurements. The following is a description of the data within the file 'Source-Data-1.xls' Each row represents data for an individual cell from strain AGY1328 (Msn2-mCherry, Dot6-GFP). There are 37 time point measurements in this table, labeled T01 through T37. Each measurement is taken 6 minutes apart (i.e T01 is at the start, T02 is 6 minutes into the experiment, T03 is 12 minutes in, and so on). T01 – T12 represent time points before NaCl is added, T13 through T22 represent the initial acute response to salt, and T23 – T37 represent the acclimated response. [Column 1] Header: cell_numb This is the unique identifier for each cell in the dataset. [Column 2] Header: colony_numb The colony_numb is different than cell_numb in that if a given cell is part of a two-cell colony, both of these cells in the colony will have the same colony_numb number, but different cell_numb numbers. For example, if two cells are part of colony 3, they will both have a 3 in their colony_numb column. For this dataset, we only analyzed colonies that were either 1 or 2 cells in size at the start of the experiment. [Column 3] Header: replicate Which of the three replicate (batch) experiments each cell comes from. [Columns 4–40] Headers: Msn2_ratio_T01 - Msn2_ratio_T37 [Columns 41–77] Headers: Dot6_ratio_T01 - Dot6_ratio_T37 Data points indicate nuclear_versus_cytoplasmic ratio (average of the top 5% of pixels divided by the median pixel intensity of all pixels in the cell) for Msn2 or Dot6, as indicated in header [Columns 78–114] Headers: col_size_T01 – col_size_T37 Each values estimates the size of the colony (measured as the number of image pixels that represent the area of a colony). [Column 115] Header: Pre_stress_growth Values represent the pre-stress growth rate, measured as the natural log of the rate of increase of colony size through the first 12 time points. [Column 116] Header: Post_stress_growth Values represent the post-stress growth rate, measured as the natural log of the rate of increase of colony size through the time points 20–29. The 36 cells removed, as described in the Methods, are NaN. [Column 117] Header: Maybe_vacuolar The 18 colonies that had apparent vacuolar localization of Msn2-mCherry during pre-stress time points are noted by '1'. All other cells are '0'.[Column 118] Header: Pre_str_Msn2_peak_sum [Column 119] Header: Pre_str_Dot6_peak_ sum These columns contain the sum of called peaks of nuclear localization for prestress time points (see *Figure 6* and Methods in main text). [Column 120] Header: Msn2_AUC_prestress [Column 121] Header: Dot6_AUC_prestress [Column 122] Header: Msn2_AUC_acclimation [Column 123] Header: Dot6_AUC_acclimation These columns are the 'Area Under the Curve (AUC) of nuclear localization' measurements as described in the Methods. [Column 124] Header: Msn2_acute_peak_height [Column 125] Header: Dot6_acute_peak_height These columns contain the 'Acute Stress Peak Height' as described in the Methods. [Column 126] Header: cycle_phase The cell-cycle phase at the moment of introduction of NaCl stress. This was called from visual inspection of the cells: presence of buds and the location of the nuclei relative to the bud where used to determine the phase: G1, S, G2, M phases. A 'q' in the column means that it was questionable what the phase was. A G0 phase is given if there is no evidence for cell division over the entire time course.

• Source data 2. Source Data 2 for AGY1813 Single Cell Measurements. The following is a description of the data within the file 'Source-Data-2.xls'. Each row represents data for an individual cell from strain AGY1813 (Msn2-mCherry, Dot6-GFP, Ctt1-iRFP). There are 45 time point measurements in this table, labeled T01 through T45 for replicates 1 and 2. There are 43 time point measurements in this table, labeled T01 through T43 for replicate 3. Each measurement is taken 6 minutes apart (i.e T01 is at the start, T02 is 6 minutes into the experiment, T03 is 12 minutes in, and so on). T01 – T12 represent time points before NaCl is added, T13 through T22 represent the initial acute response to salt, and T23 – T43/5 represent the acclimated response. [Column 1] Header: cell_numb This is the unique identifier for each cell in the dataset. [Column 2] Header: colony_numb The colony_numb is different than cell_numb in that if a given cell is part of a two-cell colony, both of these cells in the colony will have the same colony_numb number, but different cell_numb numbers. For example, if two cells are part of colony 3, they will both have a 3 in their colony_numb column. For this dataset, we only analyzed colonies that were either 1 or 2 cells in size at the start of the experiment. [Column 3] Header: replicate Which of the three replicate (batch) experiments each cell comes from. [Columns 4–48] Headers: Msn2_ratio_T01 - Msn2_ratio_T45 [Columns 49–93] Headers: Dot6_ratio_T01 - Dot6_ratio_T45 Data points indicate nuclear_versus_ cytoplasmic ratio (average of the top 5% of pixels divided by the median pixel intensity of all pixels in the cell) for Msn2 or Dot6, as indicated in header [Columns 94–138 ] Headers: Ctt1_norm_T01 – Ctt1_norm_T45 Data point indicate normalized Ctt1 abundance (median pixel intensity of all pixels

in the cell normalized by dividing the background median pixel intensity of each image that the cell belonged to). [Columns 139–183] Headers: col_size_T01 – col_size_T45 Each values estimates the size of the colony (measured as the number of image pixels that represent the area of a colony). [Column 184] Header: Pre_stress_growth Values represent the pre-stress growth rate, measured as the natural log of the rate of increase of colony size through time points T02 through T12. [Column 185] Header: Post_stress_growth Values represent the post-stress growth rate, measured as the natural log of the rate of increase of colony size through the time points 20–29. [Column 186] Header: Msn2_AUC_prestress [Column 187] Header: Dot6_AUC_prestress [Column 188] Header: Msn2_AUC_acclimation [Column 189] Header: Dot6_AUC_acclimation These columns are the 'Area Under the Curve (AUC) of nuclear localization' measurements as described in the Methods. [Column 190] Header: Msn2_acute_peak_height [Column 191] Header: Dot6_acute_peak_height These columns contain the 'Acute Stress Peak Height' as described in the Methods. [Column 192] Header: Ctt1_peak_height_max This column contains the maximum Ctt1 levels (the maximum fluorescent signal from T12 through T43 timepoint minus the median of the pre-stress T1 through T11 signal), as described in the methods. [Column 193] Header: Ctt1_time_to_cross_threshold(min) This column contains the time it took for each cell to cross a 5% change in Ctt1 abundance, as described in the methods. Cells that did not cross that threshold were not included in the timing analysis and that column was left blank for those cells.

## Data availability

All data generated and analyzed during this study are included in the manuscript and supporting files. Code used to analyze image files and generate data for cellular phenotypes is included in Source Code Files. Supplementary Data Files containing cells and associated phenotypic information are included. Source Data Files have been provided for Figure 4, and Figure 4 - supplement 1.

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
