## [Editor Report]

This paper addresses an important question in the field: the cell-to-cell heterogeneity in stress response and the functional relevance to stress adaptation. The experimental approaches are timely and their clustering and correlation analyses suggest some interesting relationships between phenotypic factors and growth adaptation.

---

## [Decision Letter]

**Decision letter after peer review:**

[Editors’ note: the authors submitted for reconsideration following the decision after peer review. What follows is the decision letter after the first round of review.]

Thank you for submitting the paper "Integrating multiple single-­cell phenotypes links stress acclimation to prior life history in yeast" for consideration at *eLife*. Your initial submission has been assessed by a Senior Editor in consultation with members of the Board of Reviewing Editors. Although the work is of interest, we regret to inform you that the findings at this stage are too preliminary for further consideration at *eLife*.

As you will see from the reviews below, the reviewers found your manuscript potentially interesting but the story incomplete (see for example reviewer #2 – "My concern is that the story seems incomplete and lacks any firm conclusions regarding causality or mechanisms. The paper relies completely on the correlation analyses, which could serve as a good starting point for a story, if followed with experimental validation (e.g. by perturbations) and mechanistic investigation. However, the authors decided to end the paper there, leaving the story incomplete and inconclusive. Therefore, a significant amount of further work will be needed to warrant publication of the paper in *eLife*."; this was pretty much the oncensus)

if you can fully address this concern through additional experiments (as well as the other concerns expressed by the reviewers), we will be happy to reconsider the paper.

*Reviewer #1 (Recommendations for the authors):*

In the present paper Bergen, Hose, McClean, and Gasch study the yeast stress response and recovery in the form of cell growth rates. It is well-known that there is great heterogeneity among the responses of otherwise genetically identical single cells to stress. An interesting question therefore is to what extent is the heterogeneity is just random stochastic noise and to what extent is it "hard-coded" into the cell based on its recent prior history (e.g. expression of various proteins, cell-to-cell variation of protein abundances, prior stress response, etc).

Clearly both – random noise and prior life history – contribute. For example, it is well known that stochastic single molecule events can regulate cell fate (Choi, Cai, Frieda, Xie, Science, 2008), but it is also well-known that if you "prime" cells by exposing them to mild stress, then they respond much better to a subsequent high-intensity stress.

An interesting question then is, what is the relative contribution of random noise and prior life history? And for prior life history, what factors are most predictive?

The setup of this paper is pretty simple. They look at two well-known factors, Msn2 and Dot6, using a single type of stress (0.7M NaCl step function) and they then quantify various aspects, the 3 most important of which are: (1) Msn2 nuclear localization; (2) Dot6 nuclear localization; and (3) cell growth rate.

The key finding is that prior Dot6 activation is more predictive than Msn2, and that a model with more variables can predict more of the variance than for example a single factor.

While the authors test multiple factors, the overall amount of the variance that can be explained is modest – around 35% using the full linear model.

I don't have major technical concerns and the work generally seems to be well done. I think the two main issues are (1) I would like to see some control computational analyses to assess how robust their growth rate quantification is and (2) the size of the dataset is pretty small, just 221 cells. This is a concern especially since they have 11 clusters, some of which have very few cells in them, raising doubt about their conclusion.

For the 1st concern, I'd like the authors to do a mock experiment: grow cells without any stress and then arbitrarily set a boundary and do similar plots to Figure 2 to see how much change in growth rate they see without stress. This will allow us to understand how much of Figure 2 is true signal and how much is just quantification noise. This could either be a new experiment or re-analysis of existing data before the stress. I'd also like to see "moving average growth rate" plots for single cells – how meaningful is a single growth rate number? I'd also like to get more detail on how growth rate was calculated. If I understand correctly, the authors use cell size. This is very reasonable, but it is a challenging quantification: since volume scales with radius to the 3rd power, tiny errors in the estimation of the radius can result in massive changes in the estimation of the volume. What was the pixel size and did they do subpixel analysis?

In general, I'd like to see a comprehensive description of how they quantified cell growth as well as a comprehensive supplementary figure "stress testing" the robustness of their quantification.

This is important since the entire paper rests on the cell growth numbers being accurate.

For my second concern, I'd like the authors to test how robust their various results are to cell numbers. I suggest that they do subsampling, where they leave out 50% of their data/cells, repeat their analysis, and then iterate multiple times to assess if all of their conclusions are robust. For example, I am concerned about whether or not they have enough data to support 11 clusters. This type of subsampling approach should be informative, though I would like to see this approach applied to all of the major conclusions and analyses.

Since both of these concerns can be assessed using either purely more computational analysis and/or just with the addition of a very simple experiment, hopefully they should not be onerous to address.

My conceptual concern is whether or not the present paper is a major conceptual advance. The notions that prior life history affects how cells deal with a stress response as well as the observation of substantial heterogeneity in single cell stress responses, are both well-known in the field and similar observations have been made in yeast and other organisms. So, the major novel contributions seem to be the relative important of Dot6 and Msn2 (Dot6 is more predictive than Msn2, which at least I did not know), and a quantification of how much one can predict from Dot6 and Msn2. This is certainly a nice contribution, and the study seems to be generally performed in a thoughtful manner, but it is perhaps a modest conceptual advance given what is already known.

*Reviewer #2 (Recommendations for the authors):*

In this manuscript, Bergen et al. combined microfluidics and time-lapse imaging to monitor single-cell phenotypes in response to acute osmotic stress. In particular, they measured translocation dynamics of two transcription factors, Msn2 and Dot6, together with a series of cellular phenotypes, e.g. cell growth, cell size, cell cycle phase, etc. To make sense of these data, they classified single cells into clusters based on their Msn2 and Dot6 dynamics, and performed correlation analyses to quantify relative contributions of measured phenotypic factors (alone or in combination) to post-stress growth rate. They found that post-stress growth rate showed a stronger correlation with an integration of multiple factors (rather than each single factor alone), among which pre-stress growth rate and Dot6 peak height seem playing major roles.

The authors focused on an important question in the field – the cell-to-cell heterogeneity in stress response and the functional relevance to stress adaptation. The experimental approaches are not new but timely. Their clustering and correlation analyses suggest some interesting relationships between phenotypic factors and growth adaptation.

My concern is that the story seems incomplete and lacks any firm conclusions regarding causality or mechanisms. The paper relies completely on the correlation analyses, which could serve as a good starting point for a story, if followed with experimental validation (e.g. by perturbations) and mechanistic investigation. However, the authors decided to end the paper there, leaving the story incomplete and inconclusive. Therefore, a significant amount of further work will be needed to warrant publication of the paper in *eLife*.

Other concerns:

1. I find the title a bit misleading. "Prior life history" sounds like a cell's previous stress encounter, nutrient condition, or age, etc. However, in the paper, it seems referring specifically to Msn2/Dot6 dynamics and pre-stress growth rate during the 72-min baseline (no stress) period immediately before the stress treatment. Maybe "pre-stress cellular state" is more accurate than "prior life history."

2. Figure S2, it seems pre-stress growth rate and Dot6 acute stress peak height are major contributing factors to post-stress growth rate. Is it possible that these two factors are mechanistically connected? Is there any correlation between these two factors? My concern here is whether these two factors can be simply combined into one factor, pre-stress growth rate or biogenesis capacity whereas Dot6 acute stress peak height simply depends on Dot6 protein expression level, reflective of cellular biogenesis capacity. If this is true, then an alternative interpretation of the correlation results will be that cell-to-cell variation in post-stress growth rate largely arises from variations in pre-stress growth rate or biogenesis capacity, which has long been known as a major source of extrinsic noise. Dot6 peak height and other phenotypic factors simply reflect pre-stress biogenesis capacity. This interpretation can also reconcile the contradiction that Dot6 peak height positively correlates with post-stress growth rate whereas Dot6 is a repressor of ribosomal biogenesis. A careful test of this possibility will be needed.

*Reviewer #3 (Recommendations for the authors):*

Bergen et al. study how the dynamic subcellular localizations of two stress-responsive transcription factors (Dot6 and Msn2) relate to variability in the growth of yeast cells before, during and after high-salt stress.

Previous studies (e.g., by the O'Shea lab) used microfluidics to look at subcellular dynamics of Msn2 and other transcription factors from a more mechanistic point of view. Other studies (e.g., Li et al. 2018 cited in the manuscript) used higher-throughput time-lapse imaging to look at population heterogeneity in growth, gene expression and stress tolerance. The present study is appealing because it sits somewhere in between. It uses microfluidics to track the two transcription factors but asks how their dynamics relate to growth.

One interesting finding, corroborating the authors' 2017 work, is that Msn2 and Dot6 do not show entirely coordinated activity. The authors identify upwards of 10 clusters of cells distinguished by different dynamic patterns of the two TFs. This finding raises the possibility that this kind of approach can find meaningful subpopulations of cells with different physiological properties.

However, the manuscript does not go far in developing understanding of how these subpopulations are generated. The authors describe Msn2 and Dot6 as both being controlled by both PKA and TOR. But that does not necessarily mean that Msn2 and Dot6 should always respond together -- one simple hypothesis is that the discordance that is seen is a result of differences in the relative contributions of these signaling pathways to Msn2 and Dot6 control. There are also counter-intuitive results that remain to be explained. In particular, the authors report that cluster 11, with below-average Dot6 response before and during stress showed slower growth. Because Dot6 represses growth-promoting genes, one would expect low Dot6 response to produce faster growth. In the end, the biggest predictor of post-stress growth rate was pre-stress growth rate, so the authors' conclusion that prior life history states are predictive (to some extent) of future ones is reasonable. But it remains to be seen why the correlation exists.

The results presented in the manuscript will certainly be of interest to those following this line of research, but the impact more generally is moderate because of the lack of deeper mechanistic insight into how this important signaling network generates heterogeneity in growth responses.

One recommendation to strengthen the presentation of the paper would be to include a figure (and/or movies) showing primary data (time-lapse images of fluorescent signal in cells in the microfluidics device), especially since this is not an off-the-shelf microfluidics platform.

Another recommendation is to flesh out more the comparisons to prior experimental work. For example, a reader new to this line of research would not immediately grasp that Li et al. 2018 followed colony growth for much longer time periods, did not use microfluidics, and used cells enriched for slow growth when examining Msn2 dynamics. These details can affect how a reader interprets the observation that Msn2 response does not correlate strongly with growth in this study. Also, the authors do not discuss any connection between growth heterogeneity and mitochondrial function, which featured heavily in the Fehrmann et al. 2013 and Li et al. 2018 papers that were cited, as well as in Dhar et al. (*eLife* 8:e38904, 2019) that was not cited.

[Editors’ note: further revisions were suggested prior to acceptance, as described below.]

Thank you for resubmitting your work entitled "Modeling single-cell phenotypes links yeast stress acclimation to transcriptional repression and pre-stress cellular states" for further consideration by *eLife*. Your revised article has been evaluated by Naama Barkai (Senior Editor) and a Reviewing Editor.

The manuscript has been improved but there are some remaining issues that need to be addressed, as outlined below:

I appreciate that the authors have put in a fair amount of work into the revision and I do believe the paper is strengthened. The authors have responded to my main concerns.

Regarding cell growth calculations, I do remain concerned. They do not do subpixel segmentation, and they only do 2D segmentation as I understand it. Furthermore, they do not quantify growth rate (which I would define as the mass accumulation per unit time). Instead, they take the relative change in the median size, and for much of the paper they then take the logarithm of this number. A lot of these assumptions seem reasonable but arbitrary and as added complications, yeast cells shrink dramatically in size upon osmotic stress and then gradually recover (so changes in cell size reflect both cell growth and osmotic changes) and budding yeast mainly grow at the bud, and the bud is frequently out-of-focus and not quantified.

Accordingly, the main output metric in the entire paper – cell growth rate – and the measured values are associated with very very large uncertainties and caveats. At a minimum, this need to be more explicitly mentioned and caveated in the main text.

Moreover, some of the analysis remains not totally convincing. For example, in Figure 5A there is a mostly random vertical scatter of points, and what seems like a fairly arbitrary straight line is drawn through it. The p-value may be small, but this does not look like a model that very well explains the data.

Overall, I appreciate the great work done by the authors to address the reviewer comments, but I do think both some technical and conceptual concerns remain. That said, it is a challenging question to tackle, and it is not clear a priori how much of the variation we should even expect Msn2 and Dot6 dynamics to explain.

---

## [Author Response]

[Editors’ note: the authors resubmitted a revised version of the paper for consideration. What follows is the authors’ response to the first round of review.]

Reviewer #1 (Recommendations for the authors):In the present paper Bergen, Hose, McClean, and Gasch study the yeast stress response and recovery in the form of cell growth rates. It is well-known that there is great heterogeneity among the responses of otherwise genetically identical single cells to stress. An interesting question therefore is to what extent is the heterogeneity is just random stochastic noise and to what extent is it "hard-coded" into the cell based on its recent prior history (e.g. expression of various proteins, cell-to-cell variation of protein abundances, prior stress response, etc).Clearly both – random noise and prior life history – contribute. For example, it is well known that stochastic single molecule events can regulate cell fate (Choi, Cai, Frieda, Xie, Science, 2008), but it is also well-known that if you "prime" cells by exposing them to mild stress, then they respond much better to a subsequent high-intensity stress.An interesting question then is, what is the relative contribution of random noise and prior life history? And for prior life history, what factors are most predictive?The setup of this paper is pretty simple. They look at two well-known factors, Msn2 and Dot6, using a single type of stress (0.7M NaCl step function) and they then quantify various aspects, the 3 most important of which are: (1) Msn2 nuclear localization; (2) Dot6 nuclear localization; and (3) cell growth rate.The key finding is that prior Dot6 activation is more predictive than Msn2, and that a model with more variables can predict more of the variance than for example a single factor.While the authors test multiple factors, the overall amount of the variance that can be explained is modest – around 35% using the full linear model.I don't have major technical concerns and the work generally seems to be well done. I think the two main issues are (1) I would like to see some control computational analyses to assess how robust their growth rate quantification is and (2) the size of the dataset is pretty small, just 221 cells. This is a concern especially since they have 11 clusters, some of which have very few cells in them, raising doubt about their conclusion.

As presented in detail below, we have added these controls and a new experiment with an additional 228 cells that validates all of our conclusions in the original manuscript.

For the 1st concern, I'd like the authors to do a mock experiment: grow cells without any stress and then arbitrarily set a boundary and do similar plots to Figure 2 to see how much change in growth rate they see without stress. This will allow us to understand how much of Figure 2 is true signal and how much is just quantification noise. This could either be a new experiment or re-analysis of existing data before the stress. I'd also like to see "moving average growth rate" plots for single cells – how meaningful is a single growth rate number? I'd also like to get more detail on how growth rate was calculated. If I understand correctly, the authors use cell size. This is very reasonable, but it is a challenging quantification: since volume scales with radius to the 3rd power, tiny errors in the estimation of the radius can result in massive changes in the estimation of the volume. What was the pixel size and did they do subpixel analysis?In general, I'd like to see a comprehensive description of how they quantified cell growth as well as a comprehensive supplementary figure "stress testing" the robustness of their quantification.This is important since the entire paper rests on the cell growth numbers being accurate.

We addressed the reviewer’s request in several ways. First, we provide additional detail on how the growth rate was calculated. The reviewer raises an important point about the challenges in calculating cell volume; indeed, small error in estimating diameters leads to large error in volume, and for this reason we did not attempt to estimate volume. Instead, we collapsed the three z-stack images and tracked the pixel area, corresponding to colony size, over time. We note that growth rate is calculated by the change in *relative size*; the median pre-stressed colony size is 4,829 pixels (where pixel size is 0.1075 x 0.1075 m) and cells changed by a median of 30% area over the course of the experiment. We did not do subpixel analysis. It is true that some calculated growth rates are under-estimated (namely, cases where the bud emerges perpendicular to the plane of analysis). However, the correlations we observe cannot be explained by this subset of cells for which growth rate is under-estimated.

Second, we now include several new analyses validating our approach (Figure 1 – supplement 2). We performed a more thorough analysis on growth rate estimates using sliding windows as requested by the reviewer. The results reveal that the growth rates are well measured and robust. (i) The logged change in cell size is very linearly correlated with time in the majority of pre-stressed cells (median R^2^ = 0.92, Figure 1 – supplement 2A). We estimated growth rates from subsets of timepoints using a sliding window as requested by the reviewer: the correlations between growth rates measured over all pre-stressed timepoints versus subsets of timepoints in a sliding temporal window is also very high for all windows (Figure 1 – supplement 2B). There is a wider distribution of linear fits post-stress (median R^2^ = 0.73, Figure 1 – supplement 2C), consistent with the range of acclimation behaviors we investigate in the manuscript. Indeed, cells that recovered growth rate after stress showed an increase in logged colony size that was well estimated by a linear fit (i.e. high R^2^), whereas cells with a lower post-stress growth rate fit less well and were more influenced by measurement noise (Figure 1 – supplement 2E), as confirmed by manual inspection. Together, these data indicate that our measurements are robust to the particular time window we investigated and that the linear fit to estimate growth rate is very high for the vast majority of cells.

Finally, we performed a mock experiment in which cells were exposed to a simple rich-media switch in the absence of NaCl stress. The vast majority of cells did not show large changes in growth rate (median ln(growth-rate change) = -0.20), compared to cells that experience the NaCl shift (median ln(growth-rate change) = -0.85, Figure 1 – supplement 2F). Together, these new experiments show that our growth rate estimates are robust and trends discussed in the manuscript are specific to NaCl stress.

For my second concern, I'd like the authors to test how robust their various results are to cell numbers. I suggest that they do subsampling, where they leave out 50% of their data/cells, repeat their analysis, and then iterate multiple times to assess if all of their conclusions are robust. For example, I am concerned about whether or not they have enough data to support 11 clusters. This type of subsampling approach should be informative, though I would like to see this approach applied to all of the major conclusions and analyses.

The revised manuscript addresses this concern in several ways. We did not intend to imply that there are precisely 11 meaningful clusters (in fact, we stated in the paper that a few of these patterns were only seen in one replicate and thus unlikely to be reproducible). We de-emphasized that there are 11 clusters in the revised manuscript. The important point here is that several of these clusters that are based entirely on transcription-factor dynamics predict pre-stress or post-stress growth rate, which were not used in the clustering. Indeed, we already showed in the manuscript that these correlations remain if we analyze cells from each replicate individually (*i.e.* subsample by replicate), as highlighted in the text.

Rather than perform additional subsampling analysis, we instead provide a better validation by adding three new replicate experiments. Although the new experiments (described more below) were done with a different strain and somewhat different microscopy settings, all of the relationships reported in the main text are recapitulated with the new data: We confirm that Dot6 peak height, but not Msn2 peak-height, is correlated with pre-stress growth rate (p = 1e-5) and post-stress growth rate (p = 0.002); modeling of post-stress growth rate is significantly influenced by Dot6 peak-height independent of its correlation with pre-stress growth rate (p = 0.04), and a multi-factorial model best explains post-stress growth rate compared to models with any single factor alone (p = 7e-6). We note that because the microscopy conditions were different, we did not attempt to merge data into a single analysis.

Finally, we show that the important clusters in Figure 4 are recapitulated in an independent mixed model clustering of the new dataset (Figure 4 – supplement 1 and Figure 5 – supplement 2). This analysis identified 6 clusters of >3 cells, all of which can be related to clusters from the original dataset. More importantly, the same subpopulation patterns originally associated with pre- and post-stress growth rate differences have the same statistically significant associations in the new dataset.

These new analyses show without a doubt that the patterns and trends we reported in the original manuscript are all valid, robust, and reproducible.

Since both of these concerns can be assessed using either purely more computational analysis and/or just with the addition of a very simple experiment, hopefully they should not be onerous to address.My conceptual concern is whether or not the present paper is a major conceptual advance. The notions that prior life history affects how cells deal with a stress response as well as the observation of substantial heterogeneity in single cell stress responses, are both well-known in the field and similar observations have been made in yeast and other organisms. So, the major novel contributions seem to be the relative important of Dot6 and Msn2 (Dot6 is more predictive than Msn2, which at least I did not know), and a quantification of how much one can predict from Dot6 and Msn2. This is certainly a nice contribution, and the study seems to be generally performed in a thoughtful manner, but it is perhaps a modest conceptual advance given what is already known.

We appreciate the concern of the reviewer and decided to add several new experiments that provide direct testing of the hypotheses put forward in the original submission. This makes more a much more satisfying study that presents important new insights into stress defense and recovery.

As this reviewer and the others point out, the connection between Dot6 and post-stress growth recovery seems counterintuitive, since a larger Dot6 stress response is predicted to produce stronger repression of growth-promoting genes. However, this result is consistent with our working model:

We previously showed that transcriptional repression by Dot6 (and its paralog Tod6) is required to redirect translational capacity away from ribosome biogenesis (RiBi) mRNAs and toward stress induced transcripts. Mutant cells lacking *DOT6* and *TOD6* fail to repress RiBi gene targets; the overabundant transcripts remain associated with ribosomes at the expense of stress-induced transcripts including Msn2-regulated *CTT1*. As a consequence, cells lacking *DOT6* and *TOD6* show delayed production of Ctt1 protein despite making more *CTT1* transcript (Ho *et al.* Current Biology, 2018).

The current manuscript expands on this model to explore why we now observe that a stronger Dot6 activation response in wild-type cells is associated with faster post-stress growth recovery. First, we show that a *dot6∆tod6∆* mutant in our culture media grows indistinguishably from wild type before stress but shows reduced growth recovery after NaCl treatment (new Figure 7A). Thus, Dot6 provides a protective response during stress and is required for normal stress acclimation.

An unresolved question is if differences in Dot6 activation in a *wild-type* cells affects Ctt1 production. We tested this here by generating a strain with three fluorescent proteins: Dot6-GFP, Msn2mCherry, and Ctt1-iRFP. This strain allows us to track growth rate, transcription factor localization dynamics, and Ctt1 production.

In fact, we find that wild-type cells with a larger Dot6 translocation peak after NaCl show higher levels and faster change of Ctt1 after NaCl treatment – this is separable from the influence of inducer Msn2: linear modeling shows that both factors contribute to Ctt1 production timing, however the contribution of Dot6 is much more significant (Msn2 activation peak is only marginally significant, p = 0.053 and explains much less of the variance in Ctt1 production timing). This is not due to association of Dot6 and pre-stress growth rate, since differences in pre-stress growth show no correlation with Ctt1production time (p = 0.21)

This is an exciting result that significantly expands our past model and our understanding of how and why cells mount a systemic response. We very much hope these extensive revisions have addressed the reviewer’s points. We believe that we have shown that our results and methods are robust, reproducible, and influential given the new insights the revised manuscript presents.

Reviewer #2 (Recommendations for the authors):In this manuscript, Bergen et al. combined microfluidics and time-lapse imaging to monitor single-cell phenotypes in response to acute osmotic stress. In particular, they measured translocation dynamics of two transcription factors, Msn2 and Dot6, together with a series of cellular phenotypes, e.g. cell growth, cell size, cell cycle phase, etc. To make sense of these data, they classified single cells into clusters based on their Msn2 and Dot6 dynamics, and performed correlation analyses to quantify relative contributions of measured phenotypic factors (alone or in combination) to post-stress growth rate. They found that post-stress growth rate showed a stronger correlation with an integration of multiple factors (rather than each single factor alone), among which pre-stress growth rate and Dot6 peak height seem playing major roles.The authors focused on an important question in the field – the cell-to-cell heterogeneity in stress response and the functional relevance to stress adaptation. The experimental approaches are not new but timely. Their clustering and correlation analyses suggest some interesting relationships between phenotypic factors and growth adaptation.My concern is that the story seems incomplete and lacks any firm conclusions regarding causality or mechanisms. The paper relies completely on the correlation analyses, which could serve as a good starting point for a story, if followed with experimental validation (e.g. by perturbations) and mechanistic investigation. However, the authors decided to end the paper there, leaving the story incomplete and inconclusive. Therefore, a significant amount of further work will be needed to warrant publication of the paper in eLife.

We agree with this reviewer (and Reviewer #1) that the original submission left some key hypotheses open ended. We hope that the revised manuscript has addressed this concern: we now provide several new experiments including perturbation analysis with a *dot6∆tod6∆* strain and new insights by following Dot6-GFP and Msn2-mCherry dynamics along with production of downstream Ctt1-iRFP protein. As outlined above and in the revised manuscript, these new insights show without a doubt that Dot6 provides a protective response during stress, is required for normal stress acclimation, and is correlated with the timing of Ctt1 induction in a way that is separable from Msn2. We believe the revised manuscript now provides important mechanistic insights that will be of broad interest and impact.

Other concerns:1. I find the title a bit misleading. "Prior life history" sounds like a cell's previous stress encounter, nutrient condition, or age, etc. However, in the paper, it seems referring specifically to Msn2/Dot6 dynamics and pre-stress growth rate during the 72-min baseline (no stress) period immediately before the stress treatment. Maybe "pre-stress cellular state" is more accurate than "prior life history."

We changed the title to address this comment and reflect the new results in the paper.

2. Figure S2, it seems pre-stress growth rate and Dot6 acute stress peak height are major contributing factors to post-stress growth rate. Is it possible that these two factors are mechanistically connected? Is there any correlation between these two factors? My concern here is whether these two factors can be simply combined into one factor, pre-stress growth rate or biogenesis capacity whereas Dot6 acute stress peak height simply depends on Dot6 protein expression level, reflective of cellular biogenesis capacity. If this is true, then an alternative interpretation of the correlation results will be that cell-to-cell variation in post-stress growth rate largely arises from variations in pre-stress growth rate or biogenesis capacity, which has long been known as a major source of extrinsic noise. Dot6 peak height and other phenotypic factors simply reflect pre-stress biogenesis capacity. This interpretation can also reconcile the contradiction that Dot6 peak height positively correlates with post-stress growth rate whereas Dot6 is a repressor of ribosomal biogenesis. A careful test of this possibility will be needed.

The reviewer raises an important point, one that we addressed in the original manuscript: although Dot6 peak height and pre-stress growth rate are partly correlated with one another, both contribute separately to explain post-stress growth rate. This is evident in the mixed linear modeling, where a model that incorporates both factors explains significantly more of the variance than either single factor model alone.

To further ensure that these factors are not simply co-variates of the same underlying feature, we added two new analyses. First, we added a principal component (PC) regression analysis. We first applied PC analysis to the four factors that were significant in the mixed linear modeling (Figure 6B). We then performed linear modeling using the PC variables and subsequently deconvoluted each PC into biological features that contribute to them. We found that 21% of the variance in post-stress growth rate is explained by PC1 + PC3, which capture the intertwined contributions of pre-stress growth rate, pre-stress Dot6 AUC, and acute-stress Dot6 peak height. It is true that these features may reflect one aspect of the cellular state, such as biosynthetic capacity (a point we present in the revised Discussion). However, an additional 14% of the variance is explained by PC4, which does not relate to pre-stress growth rate and is predominated by Dot6 behavior. Thus, Dot6 acute-stress peak height has separable explanatory power.

As an additional alternate approach, we added another analysis in Figure 6 – supplement 2: we analyzed a subset of cells that were insignificantly different from one another in pre-stress growth rate. Over this subset, there is no correlation between pre-stress and post-stress growth rate – however, there remains a significant correlation between Dot6 acute-stress peak height and post stress growth rate explaining 12% of the variance (p = 9.7e-5). Thus, without a doubt, there is a significant correlation between Dot6 behavior and the ability to recover from stress. As we report in the new manuscript, Dot6 indeed provides a protective response during stress but not before (new Figure 7) and is correlated with faster production of Msn2 target Ctt1, independent of the explanatory power of Msn2. As we expound on in the Results and Discussion, this is all consistent with and significantly expands past work from our lab showing that transient repression of growth-promoting genes is important to temporarily redirect translational capacity to induced transcripts during acclimation.

Reviewer #3 (Recommendations for the authors):Bergen et al. study how the dynamic subcellular localizations of two stress-responsive transcription factors (Dot6 and Msn2) relate to variability in the growth of yeast cells before, during and after high-salt stress.Previous studies (e.g., by the O'Shea lab) used microfluidics to look at subcellular dynamics of Msn2 and other transcription factors from a more mechanistic point of view. Other studies (e.g., Li et al. 2018 cited in the manuscript) used higher-throughput time-lapse imaging to look at population heterogeneity in growth, gene expression and stress tolerance. The present study is appealing because it sits somewhere in between. It uses microfluidics to track the two transcription factors but asks how their dynamics relate to growth.One interesting finding, corroborating the authors' 2017 work, is that Msn2 and Dot6 do not show entirely coordinated activity. The authors identify upwards of 10 clusters of cells distinguished by different dynamic patterns of the two TFs. This finding raises the possibility that this kind of approach can find meaningful subpopulations of cells with different physiological properties.However, the manuscript does not go far in developing understanding of how these subpopulations are generated. The authors describe Msn2 and Dot6 as both being controlled by both PKA and TOR. But that does not necessarily mean that Msn2 and Dot6 should always respond together -- one simple hypothesis is that the discordance that is seen is a result of differences in the relative contributions of these signaling pathways to Msn2 and Dot6 control. There are also counter-intuitive results that remain to be explained. In particular, the authors report that cluster 11, with below-average Dot6 response before and during stress showed slower growth. Because Dot6 represses growth-promoting genes, one would expect low Dot6 response to produce faster growth. In the end, the biggest predictor of post-stress growth rate was pre-stress growth rate, so the authors' conclusion that prior life history states are predictive (to some extent) of future ones is reasonable. But it remains to be seen why the correlation exists.The results presented in the manuscript will certainly be of interest to those following this line of research, but the impact more generally is moderate because of the lack of deeper mechanistic insight into how this important signaling network generates heterogeneity in growth responses.

We hope that the new experiments added to the manuscript completely address these points. Our new experiments, outlined in detail above, show that: (1) Dot6 provides a protective response during stress, since cells lacking *DOT6* and its paralog *TOD6* show wild-type growth before stress but slower growth acclimation after stress (new Figure 7A) – this is exactly consistent with results presented here, in which wild-type cells with a weaker Dot6 response show a slower growth acclimation. (2) Our past work predicts that transient Dot6-dependent transcriptional repression helps to temporarily redirect translational capacity to stress-induced transcripts. Indeed, cells lacking *DOT6* and *TOD6* show delays in producing Msn2-dependent target Ctt1. We now show here that wild-type cells with a larger Dot6 acute-stress response show statistically significantly faster Ctt1 production, separable from the contribution of Msn2 (new Figure 7). Indeed, a mixed linear model shows that the Dot6 response is the main contributor to Ctt1 production time. (3) Finally, our new analyses show that all of the patterns reported in the original manuscript are reproduced in our new datasets. We integrate these results into a new section in the Discussion that discusses the different pre-stress cellular states that these patterns may reflect.

The revised manuscripts adds important new insights that are likely to be of broad interest. With the new experiments, insights, and tested hypotheses we believe that this work will make an excellent contribution to *eLife*.

One recommendation to strengthen the presentation of the paper would be to include a figure (and/or movies) showing primary data (time-lapse images of fluorescent signal in cells in the microfluidics device), especially since this is not an off-the-shelf microfluidics platform.

We added Figure 1 – supplement 1 that shows an example cell tracked over time in the bright field, GPF, and mCherry channels.

Another recommendation is to flesh out more the comparisons to prior experimental work. For example, a reader new to this line of research would not immediately grasp that Li et al. 2018 followed colony growth for much longer time periods, did not use microfluidics, and used cells enriched for slow growth when examining Msn2 dynamics. These details can affect how a reader interprets the observation that Msn2 response does not correlate strongly with growth in this study. Also, the authors do not discuss any connection between growth heterogeneity and mitochondrial function, which featured heavily in the Fehrmann et al. 2013 and Li et al. 2018 papers that were cited, as well as in Dhar et al. (eLife 8:e38904, 2019) that was not cited.

We added several statements throughout the paper to clarify that Li et al. followed much longer time frames and that the lack of connection between stress acclimation and Msn2 response pertains to the conditions studied here. The revised Discussion provides an expanded discussion of other features that could influence the stress response, including a more direct mention of metabolic differences and mitochondrial function. We added the references suggested by the reviewer.

[Editors’ note: what follows is the authors’ response to the second round of review.]

The manuscript has been improved but there are some remaining issues that need to be addressed, as outlined below:I appreciate that the authors have put in a fair amount of work into the revision and I do believe the paper is strengthened. The authors have responded to my main concerns.

We also feel that the manuscript has been significantly improved thanks to the review process, and we thank the editor and this reviewer for the positive feedback on our revisions.

Regarding cell growth calculations, I do remain concerned. They do not do subpixel segmentation, and they only do 2D segmentation as I understand it. Furthermore, they do not quantify growth rate (which I would define as the mass accumulation per unit time). Instead, they take the relative change in the median size, and for much of the paper they then take the logarithm of this number. A lot of these assumptions seem reasonable but arbitrary and as added complications, yeast cells shrink dramatically in size upon osmotic stress and then gradually recover (so changes in cell size reflect both cell growth and osmotic changes) and budding yeast mainly grow at the bud, and the bud is frequently out-of-focus and not quantified.

The reviewer raises several points that we will address in turn. In choosing our image segmentation technique we chose to stay close to accepted methodologies in the literature (Versari, *et al.* 2017; Li, *et al.* 2018; http://yeast-image-toolkit.biosim.eu/), which do not use subpixel segmentation (see for example Bagamery, *et al.* 2020; Plavskim, *et al.* 2021; Li, *et al.* 2018; Levy, *et al.* 2012). Importantly, subpixel segmentation would add little to the accuracy of our approach: the median prestressed yeast colony size is ~4,900 pixels – as a back-of-the-envelope calculation assuming a perfect circle, this corresponds to a radius of 39.5. Extending the radius by one pixel and again assuming a simple circle for calculation, that would change the area by 5%. Thus, using subpixel segmentation would not hugely change our area estimates. The minimal error in calculating area combined with our adherence to accepted image segmentation techniques makes us confident in our analysis choices, despite some necessary assumptions with this method.

In terms of growth rate calculations: as a point of clarification, we did not use any median to calculate growth rates. We used the change in natural log of colony area in collapsed images from z-stack measurements. This is in keeping with prevailing approaches in yeast microscopy studies (e.g. Bagamery, *et al.* 2020; Levy, *et al.* 2012), which use maximum projection of colony z-stacks to define the colony area and take the linear fit as growth rate. There is a trade-off between z-resolution (*i.e.* number of z-stacks) and yeast stress induced by repeated light exposure, which limits the number of z-stacks we can take. Short of a different microscopy system, what we have done extends to the limits of our system, which we believe is accurate enough to reveal important new information, as expanded on below.

It is true that our approach will under-estimate growth rates for cells in which the bud emerges perpendicular to the slide plane; however, this noise is almost certainly leading to under-estimates of the relationship between transcription factor activation (which is well measured in all planes) to growth rate. Furthermore, many cells are well measured in our study: 57% of cells were scored as having a bud at the outset of the experiment (indicated as S, G2, or M phase, available in the data source file), and the range of pre- and post-stress growth rates as we measured them are very similar for cells scored in these different phases. Thus, while there is some noise due to budding out of field of focus, our measurements are accurate for most cells. Nonetheless, we have explicitly revised the manuscript to bring this caveat to the reader’s attention as detailed below.

The reviewer also raises concern about changes in cell volume after NaCl stress, which they argue could confound post-stress growth rate measurements. We point out that volume changes recover to pre-stress levels by 30-40 minutes (here, and also Babazadeh *et al.* 2013; Miermont *et al.* 2013), and we calculated post-stress growth rates from timepoints spanning 42 – 96 minutes after NaCl exposure in our study. Thus, we do not expect that recovery of cell volume separate from growth can explain changes in colony area at these later time points.

As an important conclusion to our arguments, we highlight that one of the major predictions made from our microfluidics and growth rate measurements – that Dot6 plays a positive role in acclimation to NaCl – is born out in bulk culture analysis that uses different methods to measure culture growth dynamics. Furthermore, the results agree with independent correlations between Dot6 activity and Ctt1 production, whose measurement is not affected by the concerns of the reviewer. Thus, the trends we report are robust and uncovering new information about stress responses, despite some necessary limitations.

Accordingly, the main output metric in the entire paper – cell growth rate – and the measured values are associated with very very large uncertainties and caveats. At a minimum, this need to be more explicitly mentioned and caveated in the main text.

As detailed in above, we feel that the noise associated with our area and growth rate measurements is reasonable and in keeping with established methodology in the field. However, we agree with the reviewer that necessary assumptions and potential caveats should be highlighted in the text. We made several changes to the manuscript to present this:

On Page 7, first paragraph of the Results: “We used the relative change in colony area over time, collapsed from multiple z-stack images per time point, as a proxy for growth rate. One limitation is that growth by this estimation will be under-estimated for cells that bud perpendicular to the slide plane, introducing noise into the growth rate measurements for some cells.”

On Page 13/14 of the Results: “We note that noise in the growth-rate measurements is likely diminishing the true fit, such that the explanatory power reported here is actually an under-estimate.”

Page 20 of the Discussion: “It will be interesting to see as technology develops for improved growth rate measurements if this fit improves further.”

On Page 24/25 of the Methods: “We note that our proxy for post-stress growth rate, taken as an indication of how well cells acclimate to salt stress, could also be influenced by differences in volume recovery for some cells, which may also be a feature of successful acclimation.”

Moreover, some of the analysis remains not totally convincing. For example, in Figure 5A there is a mostly random vertical scatter of points, and what seems like a fairly arbitrary straight line is drawn through it. The p-value may be small, but this does not look like a model that very well explains the data.

Thank you to this reviewer for making our point here, which is that the correlation shown in Figure 5A (between pre- and post-stress growth rates) *is very low*. A main point of the paper is that the correlation *improves significantly* when we incorporate other independent parameters (Figure 6B) in the fit – the point here is that the fit significantly improves from Figure 5A to Figure 6C where we explain 35% of the variance in growth rate as we measured it. As discussed above, this is likely an under-estimate due to the noise in measuring growth rate for a subset of cells, but the trends are real.

Overall, I appreciate the great work done by the authors to address the reviewer comments, but I do think both some technical and conceptual concerns remain. That said, it is a challenging question to tackle, and it is not clear a priori how much of the variation we should even expect Msn2 and Dot6 dynamics to explain.

We agree that this is a challenging problem to study but argue that our approach provides important and unexpected insights into stress biology, even if our explanatory power is under-estimated. We had already added a paragraph to the revised manuscript that discusses the numerous factors that act together to influence stress acclimation, citing other possible features not studied here. However, the relationships between stress acclimatation and Msn2 and Dot6 behavior that we report were not expected at the outset, as commented on by all three of the original reviewers, and the revised manuscript presents new models to explain these unexpected results and the role of transcriptional repression in salt acclimation. This work was just presented to world leaders in the field at the International Symposium on Fungal Stress and it was hugely well received. We believe the revised manuscript will also be very well received and make a strong contribution at *eLife.*